# Cryo-EM structure of the PlexinC1/A39R complex reveals inter-domain interactions critical for ligand-induced activation

Yi-Chun Kuo[1,4], Hua Chen[1,4], Guijun Shang[1,4], Emiko Uchikawa [2], Hui Tian[1], Xiao-Chen Bai [2,3✉] & Xuewu Zhang [1,2✉]

Plexins are receptors for semaphorins that transduce signals for regulating neuronal development and other processes. Plexins are single-pass transmembrane proteins with multiple domains in both the extracellular and intracellular regions. Semaphorin activates plexin by binding to its extracellular N-terminal Sema domain, inducing the active dimer of the plexin intracellular region. The mechanism underlying this activation process of plexin is incompletely understood. We present cryo-electron microscopic structure of full-length human PlexinC1 in complex with the viral semaphorin mimic A39R. The structure shows that A39R induces a specific dimer of PlexinC1 where the membrane-proximal domains from the two PlexinC1 protomers are placed close to each other, poised to promote the active dimer of the intracellular region. This configuration is imposed by a distinct conformation of the PlexinC1 extracellular region, stabilized by inter-domain interactions among the Sema and membrane-proximal domains. Our mutational analyses support the critical role of this conformation in PlexinC1 activation.

[1] Department of Pharmacology, University of Texas Southwestern Medical Center, Dallas, TX, USA. [2] Department of Biophysics, University of Texas Southwestern Medical Center, Dallas, TX, USA. [3] Department of Cell Biology, University of Texas Southwestern Medical Center, Dallas, TX, USA. [4]These authors contributed equally: Yi-Chun Kuo, Hua Chen, Guijun Shang. ✉email: xiaochen.bai@utsouthwestern.edu; Xuewu.zhang@utsouthwestern.edu

Semaphorin is a large family of guidance molecules that play important roles in many physiological processes, including neuronal axon guidance, cardiovascular development, and immunity[1]. Semaphorins transduce signal through the plexin receptors, which are single-pass transmembrane proteins localized on the cell surface[1–4]. The essential common downstream signaling pathway shared by all plexins is mediated by their GTPase Activating Protein (GAP) activity, which specifically catalyzes GTP hydrolysis for the small GTPase Rap and thereby converts Rap into the GDP-bound inactive state[5]. In addition, individual plexin family members can trigger unique signaling pathways by interacting with different downstream signaling proteins[2,3]. In general, these signaling pathways collectively lead to changes in cell adhesion and cytoskeleton, etc, ultimately resulting in repulsive guidance of plexin-expressing cells, including neurons, endothelial cells and immune cells[2,3].

Semaphorins and plexins are classified into seven (classes 1–7) and four classes (classes A–D), respectively, which show different specificities to one another[1]. Ectromelia virus secrets a protein named A39R that is a Sema7A mimic[6]. Semaphorins all contain an N-terminal Sema domain that adopts a 7-bladed β-propeller structure[2]. The Sema domain in most cases is followed by a plexin–semaphorin–integrin (PSI) domain, an Ig-plexin-transcription factor (IPT) domain and a C-terminal region that is highly divergent among different family members. Most semaphorins form stable dimers through the homotypic interaction of the Sema domain, which in some cases is further stabilized by the interaction between the IPT domain and an inter-chain disulfide. The extracellular region of plexins also contains an N-terminal Sema domain, a number of PSI and IPT domains[2]. The intracellular region of plexins is composed of a juxtamembrane region, the RapGAP domain and a RhoGTPase-binding domain (RBD), which are all important for signaling[7–9].

Structural analyses have shown that a dimeric semaphorin binds two plexin molecules through interactions mediated by their respective Sema domains, supporting the ligand-induced dimerization model for plexin activation[10–12]. Consistently, the RapGAP activity of the plexin intracellular region is activated upon formation of a specific active dimer[5]. Crystal structures of the zebrafish PlexinC1 intracellular region have revealed the conformation of the active dimer in both the apo and Rap-bound state[8]. These structures show that two protomers of the PlexinC1 intracellular region interact in a side-by-side mode to form the active dimer, with the two juxtamembrane regions arranged in parallel to make major contributions to the dimer interface. This conformation of the juxtamembrane region enables it to interact with the activation loop in the RapGAP domain from the dimer partner in trans, switching the RapGAP to the active conformation that can bind Rap and catalyze its GTP hydrolysis[7,8]. These observations together suggest that semaphorin-induced plexin dimer needs to have its two copies of the extracellular membrane-proximal and transmembrane regions placed close to each other in order to drive the formation of the intracellular active dimer. However, all of the reported structures of the semaphorin/plexin complexes only contain the Sema domain and one or a few following domains in plexin[10–13]. The C termini of the two truncated plexin molecules in these structures point away from each other and are separated by a large distance. Therefore, it had been unclear how this configuration of the complexes could induce the formation of the intracellular active dimer. Recent structural studies of the intact extracellular region of class A plexins in the apo state show that the three PSI domains and six IPT domains together adopt a highly curled shape, leading to an overall ring-like architecture[14,15]. Based on these structures, a docking model of the entire extracellular region of class A plexins in complex with semaphorin has been constructed, showing that the ring-

shape of plexin brings the two copies of the membrane-proximal IPT6 domain in the dimeric complex to close proximity, compatible with the formation of the intracellular active dimer[14,15]. Classes B and D plexins may use the same mechanism, as their extracellular regions have the same domain composition as class A plexins. However, the dimer of intact semaphorin/plexin complexes in the active state has not been observed directly through structural analyses to date.

The class C plexin PlexinC1 is unique in that it only has two PSI and four IPT domains in the extracellular region, rather than three and six, respectively, in classes A, B, and D plexins (Fig. 1a). The smaller number of extracellular domains in PlexinC1 suggests that it may not form the same ring-like architecture as seen for class A plexins. It is therefore not clear how the extracellular membrane-proximal domains are arranged in semaphorin-bound PlexinC1 dimer to promote the intracellular active dimer. Moreover, the transmembrane helix in plexins, which couples the extracellular and intracellular regions, remains poorly understood due to the lack of experimental structural information, although computational models based on molecular dynamics have been reported[16]. To address these questions, we purified full-length human PlexinC1 and reconstituted the 2:2 dimeric active complex with its high affinity ligand A39R from ectromelia virus. We determined the structure of the complex to 2.9 Å resolution by using cryo-electron microscopy (Cryo-EM). The structure reveals a distinct architecture of the PlexinC1 extracellular region, which is stabilized by extensive interaction among the Sema domain and the three membrane-proximal domains (IPT2–4). This architecture in combination with the particular A39R/PlexinC1 binding mode juxtaposes the two protomers of the PlexinC1–IPT4 domain in the dimeric complex properly for inducing the dimerization of the transmembrane and cytoplasmic region. The importance of these structural features for PlexinC1 signaling is confirmed by our structure-based mutational analyses. Our findings provide a structural basis for the activation of PlexinC1 and shed light on the arrangement of the transmembrane region in the ligand-induced dimer of plexins.

## Results

**Cryo-EM structure determination of the PlexinC1/A39R complex.** We expressed and purified both A39R and PlexinC1 in mammalian cells (See "Methods" for details) for cryo-EM analyses. Full-length PlexinC1 solubilized in buffers containing n-Dodecyl β-D-maltoside (DDM) formed the complex with A39R that remained intact through gel filtration chromatography (Supplementary Fig. 1), consistent with their high binding affinity as measured by isothermal titration calorimetry previously[12]. The PlexinC1/A39R complex in DDM-containing buffers however failed to yield homogenous particles in Cryo-EM images, which might be due to destabilization of the complex by the detergent. We then reconstituted the complex into peptidisc (Supplementary Fig. 1), a recently developed short peptide that can stabilize the transmembrane region of membrane proteins in the absence of detergent[17]. The sample reconstituted in the peptidisc behaved much better on EM grids and led to a 3D reconstruction of the PlexinC1/A39R dimeric complex with the C2 symmetry to an overall 3.1 Å resolution (Table 1 and Supplementary Figs. 2, 3). From the 3D classification analysis, we noticed that the angle between the two halves in the dimeric complex displayed some variations, which contributed to lower local resolution of parts of the structure that are distal to the two-fold axis. To improve the resolution, we used symmetry expansion, partial signal subtraction and focused refinement[18,19], which led to a map for one half of the dimeric complex to 2.9 Å resolution (Supplementary

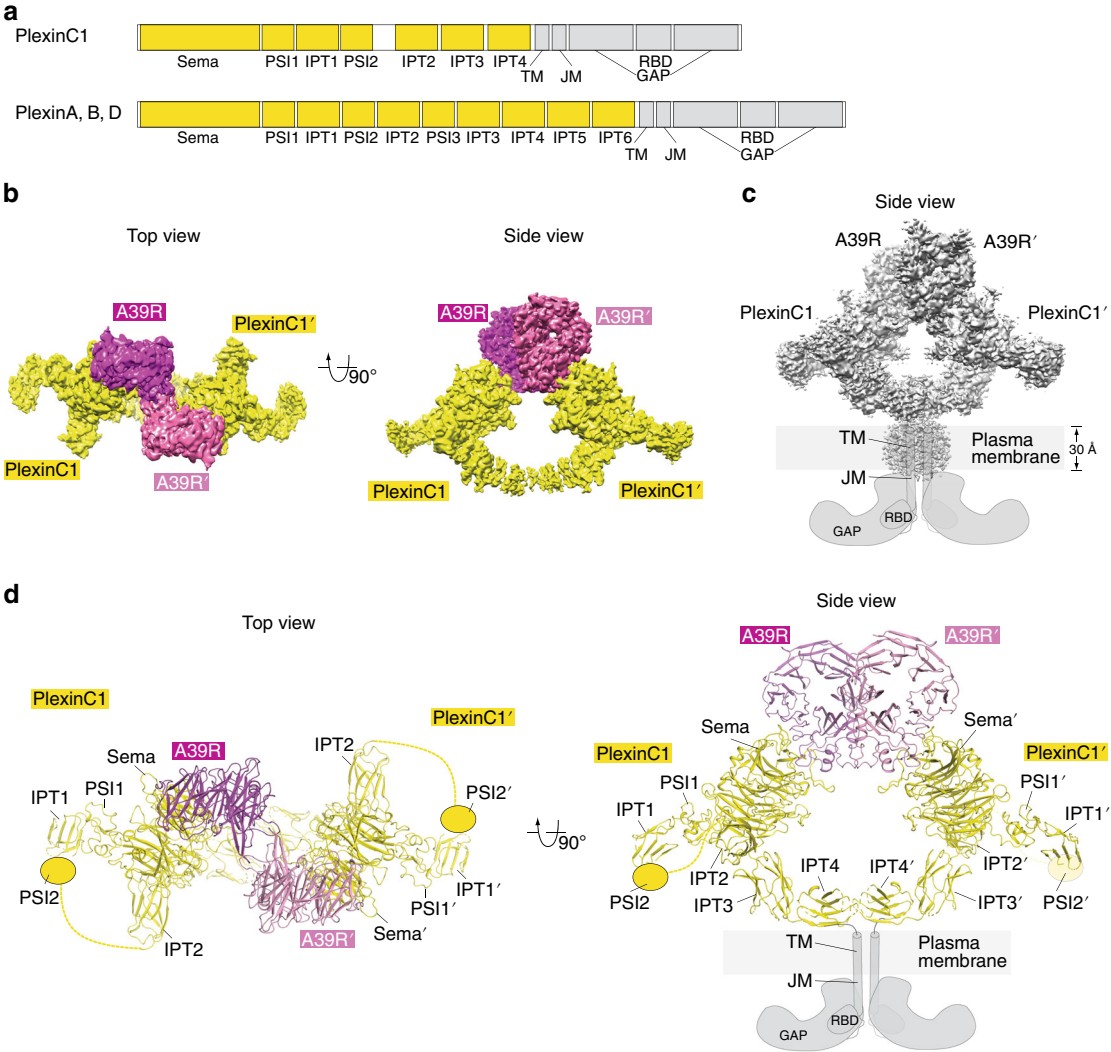

**Fig. 1 Overall structure of the full-length PlexinC1/A39R complex. a** Domain structures of plexins of different classes. IPT Ig-plexin-transcription factor domain, PSI plexin–semaphorin–integrin domain, TM transmembrane region, JM juxtamembrane region, GAP GTPase activating protein domain, RBD RhoGTPase-binding domain. **b** Cryo-EM maps of the full-length PlexinC1/A39R complex in two orthogonal views. Detailed maps of various parts of the structure are shown in Fig S4. **c** Cryo-EM map at a lower threshold showing the density of the transmembrane region of PlexinC1. Cartoon models of the transmembrane and cytoplasmic regions are drawn to indicate the overall architecture of the intact PlexinC1/A39R complex. **d** Atomic model of the dimeric PlexinC1/A39R complex. The transmembrane and cytoplasmic regions of PlexinC1 are not included in the model due to weak density.

Figs. 2–4). The map is of high quality for A39R and the extra-cellular region of PlexinC1, but weak for the transmembrane region of PlexinC1. The intracellular region of PlexinC1 is completely invisible in the cryo-EM map, presumably due to disorder as a result of the relative flexibility between the transmembrane and intracellular regions (Fig. 1 and Supplementary Fig. 2). We built an atomic model of the half complex that contains A39R, the Sema, PSI1, and IPT1–4 domains of PlexinC1 (Fig. 1). The PSI2, transmembrane and intracellular domains of PlexinC1 are not included in the atomic model. There are 3 and 24 consensus motifs of N-glycosylation in A39R and PlexinC1, respectively. Many of these sites are indeed glycosylated based on the extra densities in the cryo-EM map, and therefore partially built into the atomic model. A dimeric complex was constructed by expanding the half model according to the twofold symmetry and refined against the complete dimeric map.

**Overall architecture of the PlexinC1/A39R complex**. The cryo-EM structure of full-length PlexinC1/A39R complex shows that

the pair of the Sema domain in the A39R dimer bind two protomers of PlexinC1 at two opposite sides (Fig. 1), as seen in all the crystal structures of plexin/semaphorin complexes reported previously[10–12]. The binding interface between A39R and the Sema domain of PlexinC1 is virtually identical in the cryo-EM structure and the crystal structure of A39R in complex with the isolated Sema–PSI domains of PlexinC1 (Supplementary Fig. 5)[12]. However, the cryo-EM structure of the full-length PlexinC1/A39R complex reveals the arrangement of the extracellular membrane-proximal domains of PlexinC1 that is important for its activation by A39R.

In the previous crystal structures of ligand-bound truncated PlexinC1 and class A plexins, the Sema–PSI1 domains from the two plexin protomers are placed distal to one another (Supplementary Fig. 5). Superposition of the structures of the entire extracellular region of class A plexins with the structure of the truncated dimeric semaphorin/plexin complexes suggests that the two copies of the IPT6 domain of plexin converge near the plasma membrane as a result of the ring-shaped architecture of plexin[15]. In this model, the two IPT6 domains are roughly parallel

### Table 1 Data collection and model refinement statistics.

| | Focused refinement of the half complex | Refinement of the dimeric complex |
|---|---|---|
| Magnification | 60,386 | 60,386 |
| Voltage (kV) | 300 | 300 |
| Electron exposure (e−/Å²) | 50 | 50 |
| Defocus range (µm) | 1.5–2.5 | 1.5–2.5 |
| Pixel size (Å) | 0.828 | 0.828 |
| Symmetry imposed | C1 | C2 |
| Map resolution (Å) | 2.9 | 3.1 |
| FSC threshold | 0.143 | 0.143 |
| Initial model used (PDB code) | 3NVN | 3NVN |
| Model resolution (Å) | 2.9 | 3.1 |
| FSC threshold | 0.5 | 0.5 |
| Map sharpening B factor (Å²) | −10 | −40 |
| Model composition | | |
| Non-hydrogen atoms | 8709 | 17,418 |
| Protein residues | 1127 | 2254 |
| Ligands | 9 | 18 |
| B factors (Å²) | | |
| Protein | 69 | 100.4 |
| Ligand | 100 | 120.9 |
| R.m.s. deviations | | |
| Bond length (Å) | 0.01 | 0.012 |
| Bond angle (°) | 0.79 | 0.85 |
| Validation | | |
| Molprobity score | 2.15 | 2.93 |
| Clashscore | 12.2 | 15.0 |
| Poor rotamers (%) | 0.11 | 8.0 |
| Ramachandran plot | | |
| Favored (%) | 89.8 | 89.5 |
| Allowed (%) | 10.2 | 10.5 |
| Outliers (%) | 0 | 0 |

to each other, with their C termini pointing towards the plasma membrane. The cryo-EM structure of the full-length PlexinC1/A39R shows similar convergence of the two membrane-proximal IPT4 domains from the two PlexinC1 protomers (Fig. 1). However, due to the different architecture of its extracellular region as compared with class A plexins, the two copies of the IPT4 domain are arranged in an antiparallel fashion (See details below). The close juxtaposition of the two IPT4 protomers can promote the formation of the active dimer of the transmembrane and intracellular regions, providing the structural basis for the ligand-induced activation of PlexinC1.

**Transmembrane region of PlexinC1.** The arrangement of the IPT4 domain of PlexinC1 in the structure of the full-length PlexinC1/A39R complex suggests that the transmembrane regions of the two PlexinC1 protomers are close to each other. The transmembrane region is not included in the atomic model due to the weak density for this part in the cryo-EM map. Nevertheless, when displayed at a lower threshold, the map shows a blob of density around the expected location of the transmembrane region (Fig. 1c and Supplementary Fig. 2). The vertical length of the blob is ~30 Å, consistent with the predicted span of the ~20-residue transmembrane region of PlexinC1 adopting the α-helical conformation. The horizontal dimension is similar to the vertical span, which accommodates the two transmembrane helices of PlexinC1, the peptidisc peptides and likely some detergent and lipid molecules encompassing the transmembrane helices. It is possible that the two transmembrane helices in the dimeric structure pack closely to each other and interact directly, similar to those in the computational models of plexins and the experimental active dimers of other single-pass transmembrane receptors such as EGF receptor and the insulin receptor[16,20,21]. Such arrangement of the two transmembrane helices in the PlexinC1 dimer would be consistent with the active dimer structure of the intracellular region, where the two intracellular juxtamembrane helices following the transmembrane helix interact with each other in parallel (Figs. 1c, d)[8].

**Structure of the extracellular region of PlexinC1.** The overall architecture of PlexinC1 extracellular region appears quite different from the ring-shape of class A plexins (Fig. 2). It can be described as two curved rods arranged in a roughly orthogonal fashion (Fig. 2a). The first rod is formed by the Sema, PSI1, IPT1, and PSI2 domains, while the second is composed of IPT2–4. There is a large gap of ~30 Å between the end of the first rod and the beginning of the second (Fig. 2a). This gap is actually connected by the ~15-residue linker between the PSI2 and IPT2 domains (Supplementary Fig. 6), which is invisible in the cryo-EM map and therefore not included in the atomic model. Considering this linker between PSI2 and IPT2, PlexinC1 does have a ring-like architecture as seen in class A plexins, albeit it appears broken in the middle.

The N-terminal Sema–PSI1–IPT1 domains in both PlexinC1 and class A plexins adopt similar conformations (Fig. 2c). However, class A plexins contain one extra PSI and two extra IPT domains in the extracellular region (Fig. 1a). In addition, all the consecutive domains in class A plexins interact with one another intimately and leave no inter-domain gaps as that between PSI2 and IPT2 in PlexinC1 (Fig. 2b). The multiple domains following PSI2 in class A plexins together form a smooth curved shape to bring the membrane-proximal domains to close to the Sema domain (Fig. 2b)[15]. In one of the structures of PlexinA1 (PDB ID: 5L56), the Sema and IPT domains interact through a small interface, leading to a closed ring-shape (Fig. 2b). However, such an interface is not formed in two other structures of PlexinA1 (PDB IDs: 5L59 and 5L5C) and the structure of PlexinA4 (PDB ID: 5L5K), leaving a crack in the ring between the Sema and the membrane-proximal domains (Fig. 2b). In these cases, the position of the membrane-proximal domains in class A plexins is entirely dependent on the interactions among the consecutive domains. Small conformational differences in the nine inter-domain interfaces in the ten-domain extracellular region of class A plexins accumulate to large variations in the position of the membrane-proximal IPT6 domain in relation to the N-terminal Sema domain as seen in Fig. 2b. As a result, the distance between the two IPT6 protomers in the semaphorin-bound plexin dimers could vary substantially[15], which may affect the dimerization mode and signaling activity of the intracellular region.

In contrast to class A plexins, the PSI2 and IPT2 domains in PlexinC1 are not rigidly connected to each other due to the flexible linker between them. However, the position and orientation of the IPT2–4 domains in PlexinC1 relative to the Sema domain is locked by extensive inter-domain interactions (See details below) (Fig. 2a). Similarly, the overall conformation of the IPT2–4 rod is stabilized by many inter-domain interactions among these domains (See details below). This rigidity imposes tight conformational coupling between the ligand-binding Sema domain and the membrane-proximal domains in PlexinC1, providing the structural basis for ligand-induced dimerization and activation of its intracellular region.

The IPT domains in PlexinC1 adopt the immunoglobulin-like two-layered β-sandwich fold, with two β-sheets formed by strands A–B–E–D and C′–C–F–G, respectively (Figs. 1, 3). The overall fold of these domains is similar to that of the IPT domains in class A plexins, despite their low levels of sequence similarity (pairwise sequence identity of the multiple IPT domains between human PlexinC1 and PlexinA1 in the range of 9–25%)[15]. However, the curved rod shape of the IPT2–4 domains in PlexinC1 does not resemble the membrane-proximal IPT4–6 domains of in any of the structures of class A plexins (Fig. 2d). This distinct shape of the IPT2–4 domains together with their specific orientation relative to the Sema domain in PlexinC1 leads to close juxtaposition of the two IPT4 protomers in the dimeric PlexinC1/A39R complex (Figs. 1, 3). The two IPT4 protomers

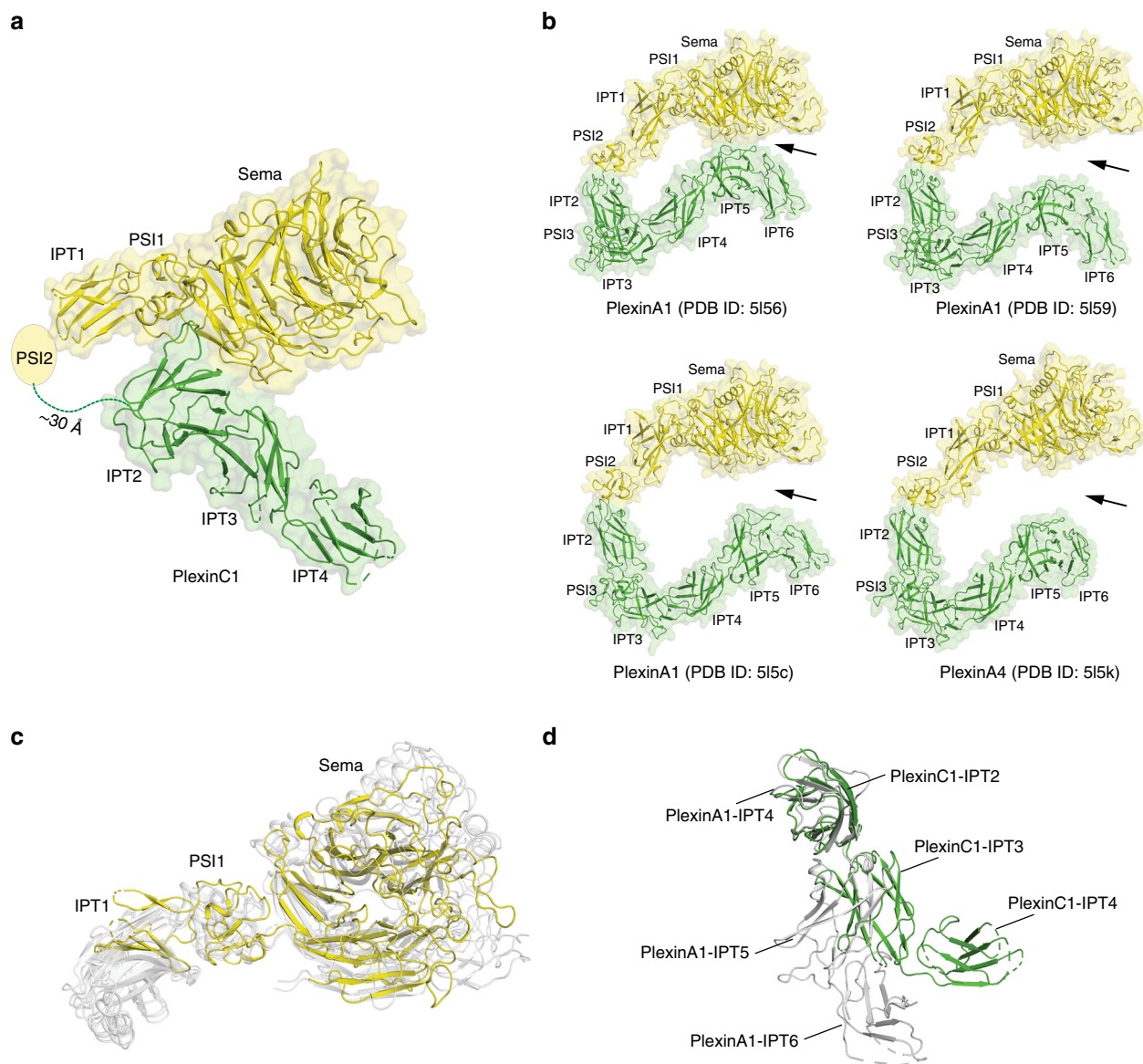

**Fig. 2 Comparison of the ectodomain structures of PlexinC1 with class A plexins. a** Overall structure of the PlexinC1 extracellular region. The Sema, PSI1, IPT1, and PSI2 together form the first rod (yellow). The IPT2–4 domains form the second rod (green) that is arranged in a roughly orthogonal orientation relative to the first rod. **b** Crystal structures of PlexinA1 and PlexinA4. The ring-shaped architecture show substantial variations among these structures, leading to large differences in the distance between the Sema and membrane-proximal domains (highlighted by arrowheads). **c** Superimposition of the Sema–PSI1–IPT1 domains of PlexinC1 (yellow) and class A plexins (gray). The superimposition is based on the Sema domain. **d** Superimposition of the three membrane-proximal IPT domains of PlexinC1 (IPT2–4) and PlexinA1 (IPT4–6) (PDB ID: 5l56). The superimposition is based on the N-terminal IPT domain of the two structures (PlexinC1–IPT2 and PlexinA1–IPT4). The conformational difference between the two structures is evident from the large deviation between the PlexinC1–IPT4 and PlexinA1–IPT6 in this superimposition.

face each other with the A–B–E–D β-sheet in an antiparallel fashion, with the β-strands roughly parallel to the plasma membrane surface (Fig. 3). In contrast, the docking models of class A plexins in complex with semaphorin suggest that the membrane-proximal IPT6 domains in the two plexin molecules are parallel to each other, with their β-strands perpendicular to the membrane surface[15].

Our structure shows that while the β-sandwiches of the two IPT4 protomers in the dimeric complex face to each other closely, they do not make any direct contact. This arrangement suggests that the architecture of the PlexinC1 extracellular region permits ligand-induced activation, but the lack of homotypic interactions in this part of PlexinC1 itself ensures no spontaneous dimerization and activation in the absence of ligand binding. The last

ordered residue of the PlexinC1–IPT4 domain in our structure is Y935 (Fig. 3). The distance between this residue and its counterpart in the dimer partner PlexinC1 is ~38 Å. The linker between this residue and the transmembrane region (nine residues from each of the two plexin molecules) (Supplementary Fig. 6) can readily span this distance and allow the two protomers of the transmembrane region to come close to each other.

**Interactions among the Sema and membrane-proximal domains.** The Sema and IPT2 domains in PlexinC1 make an extensive inter-domain interface (Figs. 2, 4). The IPT2 domain contributes the end of the β-sandwich composed on the two loops connecting β-strands C–C′ and E–F, respectively, to the interface.

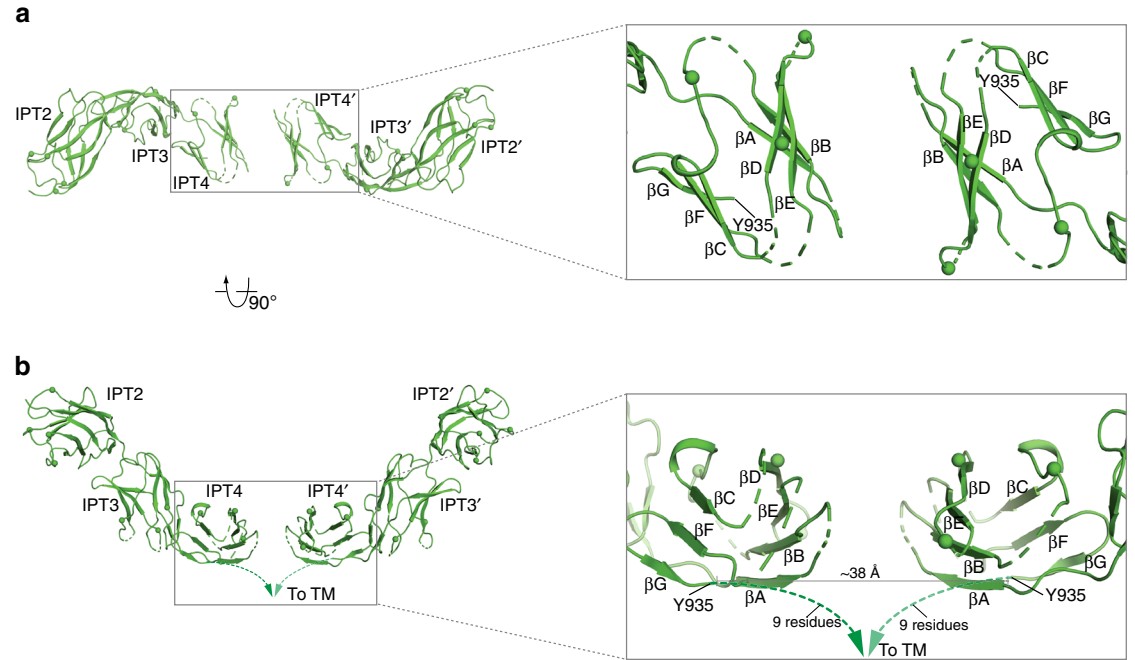

**Fig. 3 Close juxtaposition of the IPT2–4 domains in the PlexinC1/A39R complex structure.** The two protomers of the IPT2–4 domains of PlexinC1 in the dimeric PlexinC1/A39R complex are shown in two orthogonal views. The right panels show the expanded views of the two IPT4 domains, which are juxtaposed in an antiparallel fashion. Note the C′ strand in IPT4 is not modeled due to the lack of clear density. Spheres indicate the glycosylation sites. TM transmembrane region. Dashed lines indicate the 9-residue linker between IPT4 and TM.

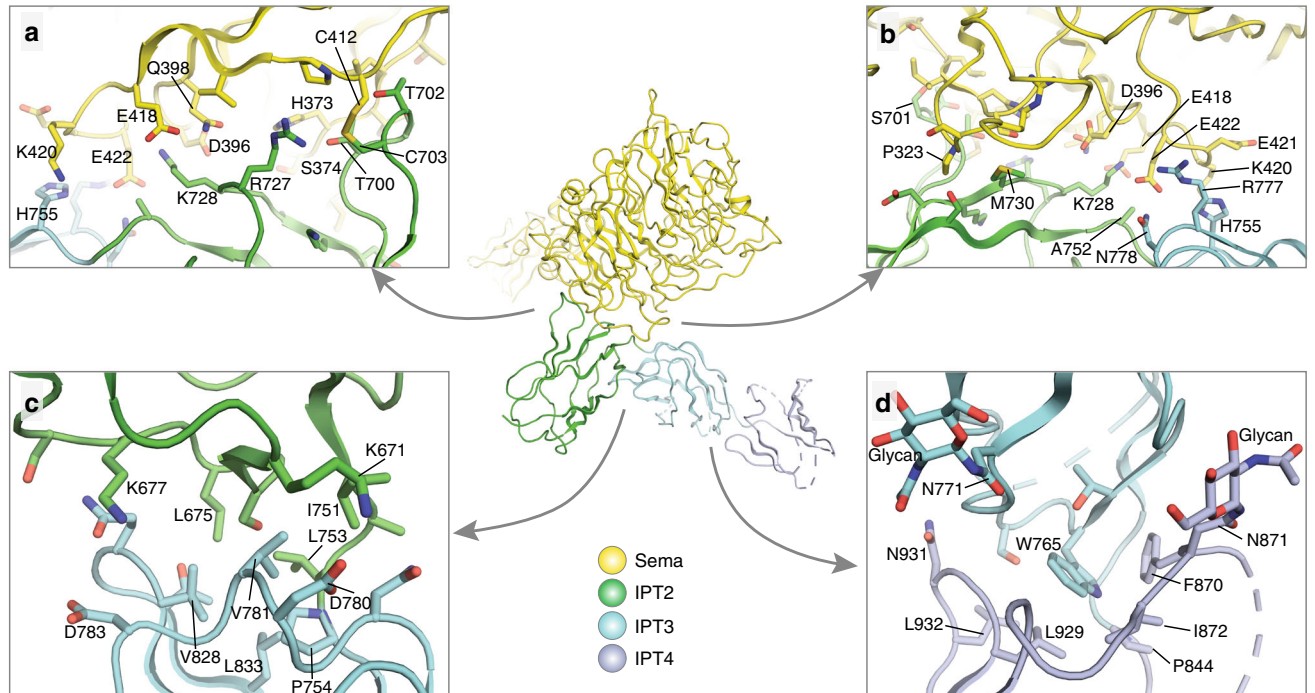

**Fig. 4 Inter-domain interactions among the Sema and IPT2–4 domains in PlexinC1. a, b** Two different views of the interactions among the Sema, IPT2, and IPT3 domains. **c** Interface between the IPT2 and IPT3 domains. **d** Interface between the IPT3 and IPT4 domains.

This part of IPT2 is embraced by the side surface of the 5th and 6th blades of the 7-bladed β-propeller in the Sema domain. In particular, Cys412 in blade-6 of the Sema domain and Cys703 in the C–C′ loop of IPT2 form a disulfide bond to covalently link the two domains (Fig. 4a). These two cysteine residues are present in PlexinC1 from various species including mammals, zebrafish, and

birds, suggesting the disulfide bond between the Sema and IPT2 domains is a conserved and functionally important feature (Supplementary Fig. 6). We further analyzed the formation of this inter-domain disulfide by expressing and purifying the extra-cellular region of human PlexinC1 with a human rhinovirus 3C protease site inserted in the linker between the PSI2 and IPT2

domains. Under reducing conditions, the protein ran as one band, which could be cleaved by the 3C protease into two fragments, corresponding to the N-terminal Sema–PSI1–IPT1–PSI2 and C-terminal IPT2–4 domains, respectively (Supplementary Fig. 7). Under nonreducing conditions, however, the protease-treated protein ran as one band with similar molecular weight as the un-cleaved protein, indicating that the N- and C-terminal fragments are linked together by a disulfide bond (Supplementary Fig. 7).

The Sema/IPT2 inter-domain interface is further augmented by many non-covalent interactions between the two domains. Thr700, Ser701, and Thr702 in the C–C′ loop of IPT2 pack against the junction between the 5th and 6th blades of the Sema domain, making numerous hydrogen bonds and van der Waals interactions (Fig. 4a, b). Two positively charged residues in the E–F loop of IPT2, Arg727, and Lys728, interact with polar residues in the 6th blade of the Sema domain, including Asp396, Gln398, Glu418, and Glu422 (Fig. 4a, b). The IPT3 domain also contributes to the interaction with the Sema domain. His755, Arg777, and Asn778 in IPT3 interact with Lys420 and Glu422 in the loop at the end of the 6th blade of the Sema domain (Fig. 4a, b). The entire interface between the Sema and IPT2–3 domains buries ~1500 Å$^2$ surface area. The large interface and the inter-domain disulfide bond together likely make the relative position between the Sema and IPT2–3 domains highly stable. In addition, the Sema/IPT2 interface is far away from the A39R-binding site on the PlexinC1 Sema domain (Fig. 1). These features together suggest that the dimerization, rather than a long-range conformation change in the extracellular region, underlies A39R-induced activation of PlexinC1. This notion is supported by the previous binding data showing that both the full-length extracellular region and the isolated Sema–PSI1 domains of PlexinC1 display monophasic association and nearly identical affinities to A39R, suggesting a simple binding mode that does not involve any allosteric conformational changes between the Sema and membrane-proximal domains in PlexinC1[12].

The IPT2 and IPT3 domains pack closely with each other because the linker between the two domains only contains two residues (Ala752 and Leu753) (Fig. 4c and Supplementary Fig. 6). The relative orientation between these two domains are restrained by many inter-domain interactions as well (Fig. 4c). The major part of the inter-domain interface is mediated by hydrophobic residues, including Leu675, Ile751 from IPT2 and Val781 and Val828 from IPT3. Lys671 and Lys677 from IPT2 form charge–charge interactions with Asp780 and Asp783 from IPT3, respectively. Similarly, the IPT3 and IPT4 domains also make an extensive inter-domain interface, with Trp765 in IPT3 surrounded by Pro844, Phe870, Ile872, Leu929, and Leu932 in IPT4 (Fig. 4d). Interesting, two glycosylation sites, on Asn771 and Asn871 in IPT3 and IPT4 respectively, are located near the IPT3/IPT4 interface and may also contribute to the stability of the inter-domain orientation (Fig. 4d). There are many additional glycosylation sites in the IPT2–4 domains (Fig. 3), which are mostly distributed away from the domain–domain interfaces and therefore are not expected to play roles in stabilizing the conformation of PlexinC1.

The inter-domain interactions described above keep the rigidity of the PlexinC1 extracellular region, particularly the membrane-proximal IPT2–4 domains, which is likely required for the proper positioning of the IPT4 domain for ligand-induced dimerization of the transmembrane and intracellular regions of PlexinC1. In support of this idea, most of the residues involved in the inter-domain interactions are highly conserved in PlexinC1 from various species, suggesting that they are functionally important (Supplementary Fig. 6).

**Inter-domain interactions in PlexinC1 critical for signaling**. To test the role of the conformation of PlexinC1 as seen in the cryo-EM structure in signaling, we designed mutations at the interface between the Sema and membrane-proximal domains. The effects on PlexinC1 signaling were assessed by using the COS-7 cell collapse assay, which reflects the ability of semaphorin-activated plexin on the surface of COS-7 cells to promote cell shrinkage, analogous to the activity of plexin in driving neuronal growth cone collapse and repulsive guidance[22]. We generated COS-7 cells stably expressing the wild type or various mutants of full-length PlexinC1. Immunostaining of A39R showed that both the wild type and mutants reached to the cell surface and bound to A39R at similar levels, which ensures that effects of the mutations was not due to misfolding or trafficking defects of the proteins in the secretory pathway (Fig. 5a).

As shown in Fig. 5b, c, ~88% COS-7 cells expressing PlexinC1 wild type underwent collapse after A39R treatment for 90 minutes. The C412A/C703A double mutation, which eliminates the disulfide bond between the Sema and IPT2 domains, decreased the cell collapse to ~10%. This result supports the notion that the inter-domain disulfide bond is essential for stabilizing the conformation of IPT2–4 in relation to the Sema domain and thereby relaying the activating signal of A39R binding to the intracellular region of PlexinC1. The D396A/E422A and K728A mutations, which disrupt the polar non-covalent interactions between the Sema and IPT2 domains, led to similar decrease of the cell collapse activity of PlexinC1. This result suggests that the non-covalent inter-domain interactions are also required for PlexinC1 signaling. During the expression and folding of PlexinC1, these non-covalent interactions may help bring the Sema and IPT2–4 domains together to promote the disulfide bond formation. In the mature protein, the disulfide bond and non-covalent interactions both contribute to the stability of the proper relative orientation between the Sema and IPT2–4 domains. As expected, the D396A/E422A/C412A/C703A mutant also displayed a very low level of collapse activity. In contrast, replacing His373 with an alanine residue only led to a modest reduction of the collapse activity, suggesting that this small nondisruptive mutation is tolerated.

We also tested whether the close juxtaposition of the two IPT4 domains in the PlexinC1 is important for signaling. We introduced a E860N/K862S double mutation to generate an N-linked glycosylation site in the βB strand of the PlexinC1–IPT4 domain, which is at the center of the A–B–E–D β-sheet that faces its counterpart in the dimeric PlexinC1/A39R structure (Fig. 3). A bulky N-linked glycan at this site is expected to obstruct the close juxtaposition of the two IPT4 protomers in the A39R-induced PlexinC1 dimer, and thereby abolish PlexinC1 signaling. As shown in Fig. 5a, this mutant was expressed on the cell surface and capable of binding A39R. However, it lost the ability to promote cell collapse upon A39R stimulation (Fig. 5b, c), supporting our model that the close proximity of the two IPT4 protomers is required for PlexinC1 activation. To further test this point, we introduced a four-residue deletion (Δ4; residues deleted: 938–941) in the linker between the IPT4 and TM of PlexinC1. Based on the structure (Fig. 3b), the remaining ten linker residues (five from each monomer) are likely insufficient to span the 38 Å distance required for the formation of the TM-intracellular active dimer of PlexinC1. The Δ4 mutant was expressed on the cell surface, albeit at lower levels compared with the wild type (Fig. 5a). Consistent with the prediction from the model, the Δ4 mutant completely failed to support A39R-induced cell collapse (Fig. 5b). Insertion of four residues (Ins4; inserted residues: Gly-Ser-Ser-Gly) in the linker between residues 939 and 940 led to a reduction in the cell collapse activity (~46% cell collapse) (Fig. 5b). These results are consistent with the model where a

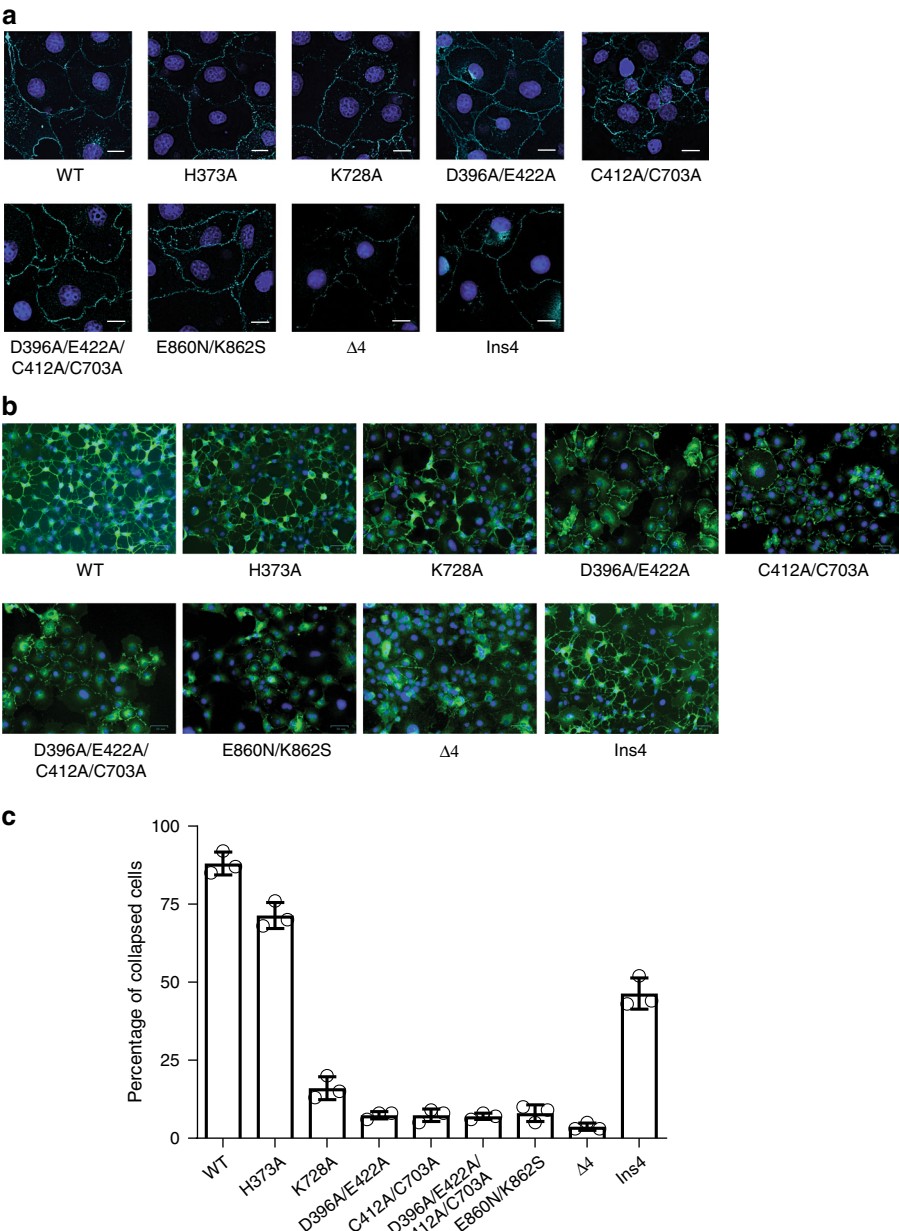

**Fig. 5 Functional analyses of the inter-domain interactions in PlexinC1 using the COS-7 cell collapse assay. a** Binding of A39R to PlexinC1 wild type or mutants expressed in COS-7 cells. Cells were treated with $His_8$-tagged A39R at 50 nM for 10 minutes, fixed and subjected to immunostaining with an anti-His-tag antibody. Cells did not undergo collapse after this short period of A39R treatment, therefore suitable for assessing cell surface expression and A39R binding of PlexinC1. A39R and the nucleus are pseudo-colored green and blue, respectively. Δ4, deletion of four residues (938–941) in the linker between the IPT4 and TM of PlexinC1. Ins4, insertion of four residues (GSSG) between residues 939 and 940. Scale bar, 15 µm. One representative image from three biological repeats is shown for each group. **b** A39R-induced collapse of COS-7 cells expressing PlexinC1 wild type or various mutants. Cells were treated A39R at 50 nM for 90 minutes. FLAG-tagged PlexinC1 was visualized through immunostaining with anti-FLAG antibody and Alexa Fluor 488-labelled anti-mouse IgG secondary antibody. PlexinC1 and the nucleus are pseudo-colored green and blue, respectively. One representative image from three biological repeats is shown for each group. **c** Quantification of the results from (**b**). The individual data points, mean and s.e.m of cell collapse percentage from three biological repeats are shown. In total, 96–147 cells in each sample were counted in each of the three biological repeats. Source data are provided as a source data file.

longer linker, while compatible with the formation of the intracellular active dimer, diminishes the coupling between the extracellular and intracellular dimers and thereby reduces the ligand-induced activation of PlexinC1.

## Discussion

The extracellular region of the plexin family members shows much lower sequence conservation than the intracellular region, which is consistent with the fact that they have different ligand specificity but share a conserved RapGAP domain for intracellular signaling[7,8,10]. The high degree of conservation of the RapGAP suggests that the activation mechanism as shown by the active dimer structure of zebrafish PlexinC1 intracellular region is representative of all plexin family members[5,8]. However, the extracellular region of PlexinC1 contains three fewer domains compared with plexins of classes A, B, and D, indicating different

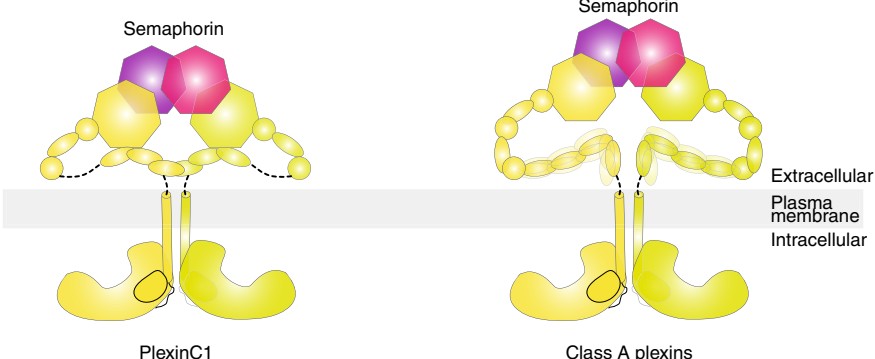

**Fig. 6 Cartoon models of the active complexes of PlexinC1 and class A plexins.** The models highlight the different architecture between PlexinC1 and class A plexins. The conformation of the membrane-proximal IPT2–4 domains in PlexinC1 is stable due to their tight interactions with the Sema domain. In contrast, previous crystallographic studies have suggested that the conformation of the membrane-proximal domains in class A plexins show substantial variations (indicated by the blurring effect). Structures of full-length class A plexins in complex with semaphorin are not available. The model of class A plexins is constructed by combining the structures of the apo state intact PlexinA ectodomains and the structures of the N-terminal ligand-binding domain of PlexinA in complex with semaphorin.

structures and ligand-induced dimerization modes. The cryo-EM structure presented here explains how this distinct structure of PlexinC1 places the membrane-proximal domains properly to allow the dimerization of the transmembrane and intracellular regions (Fig. 6). The model of class A plexins bound to their ligands suggests that while the structures of the extracellular region of class A plexins are different, the membrane-proximal domains are placed close to the ligand-binding Sema domain (Fig. 6)[15]. These analyses together therefore lead to a unifying theme in the activation of different plexin family members by their respective ligands, despite the sequence and structural divergence. Nevertheless, different orientations of the most membrane-proximal domains in PlexinC1 (IPT4) and class A plexins (IPT6) suggest that the C termini in the different plexins are related to their dimer partner differently (Fig. 6). The effects of this difference on the dimerization of the transmembrane and intracellular region are not clear at present. The length of the linker between this domain and the transmembrane region is predicted to be short (~5–10 residues among different plexin family members). The different arrangement may be fine-tuned to adapt to the various linker lengths and structures to allow optimal activation of the intracellular region.

Class A plexins in the apo state have been shown to form an inhibitory dimer, which plays a role in preventing spontaneous activation of the receptor in the absence of ligand binding[15,23]. The interface is formed between the Sema domain of one plexin molecule with the PSI2–IPT2 domains of the second plexin molecule in the dimer. The extracellular region of PlexinC1 in the absence of ligand binding has been indicated to form oligomers[12]. However, given the very different domain structure and conformation, it is unclear whether PlexinC1 forms an inhibitory dimer similar to that of class A plexins. Further studies are required to address the existence and nature of the inhibitory dimer of PlexinC1. Another interesting aspect of class A plexins is that, in addition to the ring-like structure, they may adopt a twisted-open chair-like conformation as shown by negative-stain EM images, which may offer an additional layer of regulation of their activity[15]. In contrast, the disulfide bond between the Sema and IPT2 domains of PlexinC1 renders such a twisted conformation inaccessible to PlexinC1.

There is mounting evidence suggesting that the transmembrane region of single-pass transmembrane receptors such as plexins, insulin receptor and the EGF receptor makes specific interactions that are important in regulating receptor activity, rather than merely acting as a passive linker between the extracellular and intracellular regions[20,21,24–26]. The coupling mechanisms of the extracellular and intracellular regions by the transmembrane region in these receptors is poorly understood because structural analyses of full-length single-pass transmembrane proteins remain challenging. None of the few reported structures of full-length single-pass transmembrane receptors has the transmembrane region resolved to high resolution. The transmembrane region of PlexinC1 in the cryo-EM structure presented here is also not well resolved, although weak density is present. In addition, it remains possible that the peptidisc might have imposed some restraints on the conformation of the PlexinC1 transmembrane region. The sequences of the transmembrane region of PlexinC1 and other plexin family members are hydrophobic characteristic of transmembrane helices, but show no obvious conserved motifs for mediating the dimerization. Recent studies have shown that transmembrane helices can dimerize through surprisingly diverse interactions[26], which makes sequence-based prediction of the dimerization modes difficult. Therefore, further optimization of the peptidisc or other types of nanodiscs together with the presence of lipids are needed for better mimicking the plasma membrane environment that stabilizes the transmembrane region of PlexinC1, allowing the determination of its high-resolution structure. Such structures of full-length PlexinC1 and other single-pass transmembrane receptors with all domains resolved at high resolution will greatly enhance our understanding how the transmembrane region relays the ligand-induced dimerization of the extracellular region to induce the specific active dimer of the intracellular region.

## Methods

**Protein expression and purification**. The cDNA of A39R (residues 15–399; UniProtKB, Q8JL80) was synthesized by Integrated DNA Technologies and cloned into a modified pEZT-BM vector[27] that encodes an N-terminal heterologous signal peptide (MGILPSPGMPALLSLVSLLSVLLMGCVAE) and a C-terminal His$_8$-tag. All primers used in the study are listed in Supplementary Table 1. The plasmid was transfected to Expi293F cells (ThermoFisher, #14527) by using polyethylenimine (PEI) as the transfection reagent. Valproic acid and sodium propionate were added to Expi293F cells to enhance protein expression. Six days after transfection, cells were spun down. The supernatant was collected, concentrated, and diluted with at least 10X volume of buffer A containing 10 mM Tris (pH 8.0), 500 mM NaCl, 5% glycerol (v/v), and 20 mM imidazole. Buffer diluted supernatant was passed through a 1 ml HisTrap column (GE Healthcare) to capture the A39R protein. After washing with buffer A, the A39R protein was eluted by a buffer containing 10 mM Tris (pH 8.0), 500 mM NaCl, 5% glycerol (v/v), and 250 mM imidazole. The protein was further purified by a Superose 6 10/30 gel filtration column (GE healthcare) equilibrated with a buffer containing 20 mM HEPES (pH 7.5) and 150 mM NaCl. Purified protein was concentrated and stored at −80 °C.

The cDNA of human PlexinC1 was from ThermoFisher (UT Southwestern human cDNA library). The coding region of the full-length PlexinC1 (residues 1–1568) fused to a cleavage site for the human rhinovirus 3C protease, T6SS secreted immunity protein 3 (Tsi3) from *Pseudomonas aeruginosa*[28] and a His$_8$-tag at the C terminus in tandem was inserted into the pEZT-BM vector. The plasmid was used to generate recombinant baculovirus for protein expression in HEK293 GnTI$^-$ cells (ATCC, #CRL-3022) with the BacMam system[29]. GnTI$^-$ cells were then infected by the baculovirus at 1:50 (v/v) ratio. Sodium butyrate at 5 mM was supplemented 24 hrs after infection to enhance protein expression. Ninety-six hours after infection, cells were pelleted and resuspended in a lysis buffer containing 20 mM HEPES (pH 7.5), 400 mM NaCl, 1 mM CaCl$_2$, DNase I, serine protease inhibitor AEBSF (Sigma, #A8456), and a protease inhibitor cocktail (Sigma, P8340). Cells were lysed by a cell disruptor and centrifuged at 35,000 rpm for 1 h at 4 °C. Membrane pellet was resuspended by a homogenizer in the lysis buffer without DNase I. Membrane proteins were solubilized by stirring at 4 °C overnight after adding 2% n-dodecyl-B-d-maltopyranoside (DDM, Anatrace, #D310) and 0.4% cholesteryl hemisuccinate tris salt (CHS Anatrace,#CH210). The mixture was centrifuged at 45,000 rpm for 1 h at 4 °C. The supernatant was cleared by passing through 0.45 µM nylon membrane. Purification of Tsi3-tagged PlexinC1 was carried out in a similar manner as previously described[30]. Briefly, the protein was captured by Sepharose 4B resin (GE Healthcare) conjugated with the T6SS effector protein Tse3 equilibrated in buffer B containing 20 mM HEPES (pH 7.5), 400 mM NaCl, 1 mM CaCl$_2$, 0.05% DDM, and 0.01% CHS. After thorough washing with buffer B, the PlexinC1 protein was cleaved from the Tsi3-tag by incubating overnight with His$_6$-tagged rhinovirus 3C protease in the presence of 25 mM imidazole. The elutant was passed through Ni-NTA Sepharose resin (GE Healthcare) equilibrated with buffer B containing 25 mM imidazole to trap the Tsi3-His$_8$-tag and His$_6$-tagged protease. The flow-through was further purified by a Superdex 200 Increase 10/300 GL gel filtration column (GE healthcare) equilibrated with a buffer containing 20 mM HEPES (pH 7.5), 150 mM NaCl, and 0.03% DDM. Purified protein was concentrated and stored at −80 °C.

The extracellular region of human PlexinC1 (residues 1–942) was cloned into the pRK5 vector (BD Biosciences) with a C-terminal His$_8$ tag. The construct contains a segment that encodes a human rhinovirus 3C protease cleavage site inserted between Glu649 and Asn650 of PlexinC1, introduced by polymerase chain reactions. The protein was expressed in Expi293F cells with PEI transient transfection. The purification procedure was the same as that for A39R. The purified protein was concentrated to 3.3 mg/ml in the buffer containing 20 mM HEPES (pH 7.5) and 150 mM NaCl.

**Gel analyses of the human PlexinC1 extracellular region.** Protease digestion of the PlexinC1 extracellular region containing the 3C protease cleavage site in the linker was carried out by incubating the protein (10 µg) with sumo-tagged 3C protease (0.1 µg) in a 10 µl reaction system at room temperature for 1 h. Protein samples were mixed with loading dye with or without dithiothreitol and subjected to electrophoresis on a 4–20% gradient gel (Bio-rad).

**Reconstitution of the PlexinC1/A39R with peptidisc.** The peptidisc peptide was purchased from PEPTIDISC LAB (https://peptidisc.com). The sequence of peptidisc peptide is FAEKFKEAVKDYFAKFWDPAAEKLKEAVKDYFAKLWD (N-terminus to C-terminus, single-letter amino acid code)[17]. The peptidisc peptide powder was dissolved at 2 mg/ml using 20 mM HEPES pH 7.5 and 150 mM NaCl. To help solubilization, the peptidisc solution was incubated at 50 °C water bath for 20 min and vortexed. Reconstitution of the PlexinC1/A39R complex into peptidisc was carried out by following the protocol as described in[17]. Briefly, 300 µg of PlexinC1 was mixed with A39R at 1:1 molar ratio and incubated for 10 min at 25 °C. 60X molar excess of peptidisc peptide (460 µg) was then added to the PlexinC1/A39R mixture and incubated at 4 °C with constant rotation overnight. The mixture was centrifuged at 15,000 rpm for 10 min at 4 °C to remove precipitated protein and then loaded to a Superose 6 Increase 10/300 GL column equilibrated with a buffer containing 20 mM HEPES (pH 7.5) and 150 mM NaCl. Fractions containing the PlexinC1/A39R complex were pooled, concentrated to ~0.5 mg/ml, and stored at −80 °C.

**Cos-7 cell collapse assay.** FLAG-tagged PlexinC1 wild type or mutants were cloned into a modified pLVX vector (Takara Bio., #632183). Lentivirus were produced by transfection of HEK293T cells with the plexin constructs and packing plasmids. Supernatants containing viral particles were filtered with 0.45 µm syringe filter and concentrated by centrifuging at 26,000 rpm for 2 h. After centrifugation, the supernatant was aspirated and viruses were resuspended in PBS. To generate COS-7 cells stably expressing PlexinC1, COS-7 cells (ATCC, #CRL-1651) were infected with the virus and selected with 5 µg/ml puromycin. To visualize the binding of A39R by PlexinC1 expressed on the cell surface, cells were incubated with His$_8$-tagged A39R at 50 nM for 10 min at room temperature. Unbounded A39R was washed away with PBS. Cells were fixed with 4% PFA and blocked with 5% bovine serum albumin. An anti-His antibody (Takara Bio., #631212; 5000X dilution) and Alexa-488-coupled anti-mouse IgG secondary antibody (Thermo-Fisher, A11029; 1000X dilution) were used to stain His$_8$-tagged A39R. Nuclei were stained with DAPI (4,6-diamidion-2-phenylindole). Images were taken by a

DeltaVision Core microscope and processed with deconvolution and Z-stack quick projection using the SoftWoRx 7.0 software package provided by DeltaVision. To test A39R-induced COS-7 cell collapse, cells expressing PlexinC1 wild type or mutants were seeded in four-well cell culture chamber slide (ThermoFisher, #154526) at 50,000 cell/well. Twenty-four hours after seeding, cells were washed three times with HBHA buffer (150 mM NaCl, 20 mM HEPES, pH 7.0, 1 mM MgCl$_2$, 5 mM CaCl$_2$, and 0.05% BSA). Cells were then treated with A39R at a final concentration of 50 nM in HBHA buffer for 90 minutes to induce collapse. For better visualizing of the morphology, cells were stained for PlexinC1 and nuclei with an anti-FLAG antibody (Sigma, F1804; 3000X dilution) and DAPI, respectively. Images were acquired at 200X magnification with a ZOE fluorescent cell imager (Bio-Rad). Collapsed cells were identified based on substantially reduced area and star-like morphology as compared with the characteristic large and round shape of normal COS-7 cells[31]. Counting of cells were carried out in a blinded manner. Data were plotted in GraphPad Prism 8.

**Cryo-EM data collection.** The A39R/PlexinC1 complex in peptidiscs at 0.5 mg/ml was applied to a glow-discharged Quantifoil R1.2/1.3 300-mesh gold holey carbon grid (Quantifoil, Micro Tools GmbH, Germany), blotted under 100% humidity at 4 °C and plunged into liquid ethane using a Mark IV Vitrobot (FEI). Micrographs were collected on a Titan Krios microscope (FEI) with a K3 Summit direct electron detector (Gatan) operated at 300 kV using the SerialEM software[32]. The GIF-Quantum energy filter was set to a slit width of 20 eV. Images were recorded in the super-resolution counting mode with the pixel size of 0.828 Å. Micrographs were dose-fractioned into 32 frames with the dose rate of 1.5 e$^-$/Å/frame.

**Image processing and 3D reconstruction.** The procedure of image processing and 3D reconstruction is outlined in Supplementary Fig. 2. Motioncorr2 1.1 was used to perform twofold binning, motion collection and dose weighting of the movie frames[33]. CTF correction were carried out using GCTF 1.06[34]. The following image processing steps were carried out in RELION 3.0 or 3.1[35]. Auto-picked particles were extracted and binned by 3 or 4 times and used in 2D classification. Particles in good 2D classes were selected and subjected to 3D classification using an initial model generated from a subset of the particles. A total of 184,225 particles from two good 3D classes were re-extracted to the original pixel size. 3D refinement of these particles with the C2 symmetry imposed led to a 3D reconstruction to an overall resolution of 3.4 Å, although the transmembrane and intracellular region of PlexinC1 were completely missing in the density map. Based on this reconstruction, two strategies were used to improve the density for the membrane-proximal and transmembrane domains. In the first, the particles of the initial 3D reconstruction were subjected to 3D classification with local angular search. One class (containing 27,967 particles) that showed stronger density for the membrane-proximal and transmembrane region was chosen for the final 3D refinement and postprocessing. The resolution of the resulting 3D reconstruction was similar to the previous one, but had better local resolution for the membrane-proximal region and show weak density for the transmembrane region (Fig. 1c and Supplementary Fig. 2c, lower left panel). In the second strategy, the particles of the initial 3D reconstruction were subjected to CTF refinement and high-order aberration correction in Relion 3.1, and then partial signal subtraction to keep only the membrane-proximal and transmembrane region[18]. 3D classification of these partial-subtracted particles without alignment led to one class that showed much better density for the membrane-proximal domains of PlexinC1. The particles in this class were reverted to un-subtracted version and used in 3D refinement with the C2 symmetry. The resulting 3D reconstruction were used as the basis for twofold symmetry expansion of the dataset, and partial subtraction of half of the dimeric A39R/PlexinC1 complex[18,19]. 3D refinement and postprocessing of the half complex led to a reconstruction of 3.0 Å resolution. The rationale behind this strategy was that the two halves of the dimeric complex appeared to have some hinge motion around the center dimer interface of A39R where the twofold symmetry axis passes, leading to poorer alignment of parts that are far away from A39R/the twofold axis. Symmetry expansion and partial signal subtraction allowed each half of the dimeric complex to be better aligned, which reduced the detrimental effects of the motion. Another round of 3D classification without alignment focused on the membrane-proximal domains and subsequent 3D refinement and postprocessing generated a 3D reconstruction of 2.9 Å resolution, with the local resolution of the membrane-proximal domains substantially improved. Resolution was estimated by applying a soft mask around the protein density with the Fourier Shell Correlation (FSC) 0.143 criterion. Local resolution was calculated in Relion.

**Model building and refinement and validation.** The 3D reconstruction of the half complex from the focused refinement shows excellent density for A39R, the Sema, PSI1, and IPT2 domains of PlexinC1 for model building (Supplementary Fig. 4). The density for the IPT1, PSI2, IPT3, and IPT4 domains of PlexinC1 were weaker, and therefore only partially built in the model (Supplementary Fig. 4). The transmembrane and cytoplasmic domains of PlexinC1 are very weak in this map. The following residues are not included in the final model due to the lack of density: 392–399 (A39R); 33–36 (PlexinC1-N-terminal); 512–519, 537–543, 550–555, 559–572 (PlexinC1-IPT1); 587–658 (PlexinC1-IPT1-PSI1-linker); 790–792, 814–819 (PlexinC1–IPT3); 851–856, 865–868, 882–887, 896–903,

911–923 (PlexinC1–IPT4), and 936-end (PlexinC1–IPT4-end). Sidechains of many residues in the less well-ordered domains are also omitted. Model building was started by docking the crystal structure of A39R in complex with the Sema–PSI domain of PlexinC1 (PDB ID: 3NVN)[12]. Additional domains were manually built in Coot 0.9[36]. Determination of sequence register of the IPT domains in PlexinC1 was facilitated by secondary structure predications, comparisons with the IPT domains of class A plexins, bulky residues, and glycosylation groups. Real-space refinement of the model against the map was carried out in the Phenix package (V1.7), with secondary structure restraints and Ramachandran restraints[37]. Two copies of the half complex were docked into the map of the intact complex and subjected to real-space refinement with twofold symmetry constraints to generate the final 2:2 PlexinC1/A39R complex. Model validation were carried out by using MolProbity as a part of the Phenix validation tools (Table 1)[38]. Figures were generated in PyMOL 2.3 (Schrödinger, LLC, 2015, the PyMOL Molecular Graphics System) or Chimera 1.3[39].

**Reporting summary**. Further information on research design is available in the Nature Research Reporting Summary linked to this article.

## Data availability

Data supporting the findings of this manuscript are available from the corresponding authors upon reasonable request. A reporting summary for this article is available as a Supplementary Information file.

The Cryo-EM density maps of the PlexinC1/A39R complex have been deposited in the Electron Microscopy Data Bank (accession code: EMD-21442). The atomic coordinates for the complex have been deposited to the RCSB Protein Data Bank, PDB 6VXK [10.2210/pdb6VXK/pdb]. The source data underlying Fig. 5c are provided as a Source Data file.

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

## Acknowledgements

Cryo-EM data were collected at the University of Texas Southwestern Medical Center (UTSW) Cryo-Electron Microscopy Facility that is funded by the Cancer Prevention and Research Institute of Texas (CPRIT) Core Facility Support Award RP170644. We thank Drs Nicastro and Stoddard for facility access and data acquisition. We thank Hongtao Yu and the Structural Biology Lab at UTSW for equipment usage. This work is supported in part by grants from the National Institutes Health (R35GM130289 to X.Z. and R01GM136976 to X.-C.B.), the Welch foundation (I-1702 to X.Z. and I-1944 to X.-C.B.) and CPRIT (RP160082 to X.-C.B.). X.-C.B. and X.Z. are Virginia Murchison Linthicum Scholars in Medical Research at UTSW.

## Author contributions

X.Z., X.-C.B., and G.S. conceived the project. Y.C.K., H.C., G.S., and H.T. optimized the protein expression and purification procedures and prepared samples for cryo-EM analyses. H.C., Y.C.K., and G.S. did the COS-7 cell collapse assay. X.-C.B., E.U., and X.Z. collected the cryo-EM data. X.-C.B and X.Z. carried out Cryo-EM reconstruction and model building. X.Z., X.-C.B., Y.C.K., and H.C. wrote the paper with inputs from other authors.

## Competing interests

The authors declare no competing interests.
