## [Peer Review File · Nature Communications]

Reviewers' comments:

Reviewer #1 (Remarks to the Author):

Plexin family proteins represent major class of signaling receptors for semaphorin ligands, which play important roles in various cellular events implicated in normal development as well as diseases. Plexin-mediated signaling requires activation of its GAP activity located in the intracellular region, generally believed to be accomplished by dimerization upon the engagement of dimeric semaphorin ligand at the extracellular face. However, current "plexin activation model" lacks structural and mechanistic precision due to the fact that it is based on the structural information obtained for various partial fragments of both ligands and receptors, as in the case for virtually all single-pass membrane receptor systems.

In this paper, Kuo et al made a huge step toward the complete understanding of the plexin activation on the membrane, by reporting a 3.1Å cryo-EM structure of the entire plexinC1 polypeptide in complex with a semaphorin surrogate ligand A39R. Although they could not build models for the TM and the cytoplasmic GAP domains, they succeeded in visualizing nearly all ectodomain regions for plexin C1 and bound ligand, presenting the global architecture of the 2:2 plexin-ligand complex formed at the extracellular side for the first time. The structure revealed that the two TM domains are indeed in close proximity, supporting the hypothesis that the ligand-induced ectodomain dimerization of plexin leads to the GAP domain dimerization. They also discovered unexpected intra-molecular interface between the sema domain and the "bottom" of the IPT2 domain, which makes the plexin C1 ectodomain highly bent and compact, despite the presence of several Ig-like modules in its "stalk". Apart from the new information related to the plexin signaling mechanisms, this paper also offers useful information regarding the important technical advancement to the field, which is the use of peptidisc approach. Despite extensive efforts by numerous experts in the field, we still do not have atomic resolution structures of single-pass membrane proteins with large ectodomain. This is in contrast to the multi-pass membrane proteins, of which high resolution structures by either cryo-EM or crystallography are reported almost weekly. Peptidisc approach is much less popular than other reconstitution methodologies including nanodisc, so the result by Kuo et al may spark new wave of investigation of single-TM protein reconstitution by using this relatively new technology. Overall, I think this fine paper should be published without unnecessary delays. There are few specific comments/requests I would like to make.

1. Disulfide bridge connecting Cys412 and Cys703

I have a concern about the authenticity of this disulfide. Although this assignment might be consistent with the sequence conservation pattern among different plexins (should be nice to have sequence alignment figure here), the authors do not show experimental evidence to back up this notion. At least very clear density map near this disulfide must be shown, but even then the conclusion must be drawn very carefully considering the low resolution of the data. My concern stems from the puzzling data that charge-neutralizing mutations at the interface alone (i.e., D396A/E422A and K728A) can lead to the disrupted signaling phenotype (Fig. 5). The C412-C703 disulfide would covalently fix the sema-IPT2 interface and hence any mutations at the interface (particularly when they are Ala mutations) should not be able to affect the conformation. The authenticity of this kind of long-range disulfide bond is extremely important to correctly understand the conformational regulation of a receptor, so I urge authors to address this point more rigorously. A limited protease digestion on the full-length (or recombinant soluble ectodomain) plexinC1 could clarify this point, because the disordered PSI2-IPT2 linker region should be susceptible to such digestion. If authors cannot give more supporting evidence for the disulfide, they should mention that the disulfide formation is one of the potential explanations among other possibilities.

2. IPT domain conformations and arrangements

The authors state that "the curved rod shape of the IPT2-4 in PlexinC1 does not resemble any of the three consecutive IPT domains in class A plexins". This notion is central to their point of argument regarding the unique IPT domain conformation (and signaling mechanism) of PlexinC1, but unfortunately the current figures cannot convey this message. I suggest authors to make superpositions of the consecutive as well as individual IPT domains to help readers to appreciate similarity and difference among various IPT domains. Also, a supplementary figure showing

structurally-aligned IPT domain sequences including other plexin classes, with secondary structure elements clearly labeled, would be very informative to readers.

3. peptidisc reconstitution

In general, the method section is written with sufficient details. The EM analysis part is well performed, applying some of the analytical approaches that are at the forefront of the field. Unfortunately, the authors give only limited information regarding their preparation of the peptidisc reconstitution, which is probably the most wanted information from the membrane protein researchers. I urge authors to expand this section and describe fully the content of the peptidisc with actual sequence of the peptide used, so that readers do not have to go back and forth between the ref 15 and the company's URL.
(end of comments)

Reviewer #2 (Remarks to the Author):

The paper by Bai, Zhang and coworkers claims to reveal inter-domain interactions critical for the ligand induced interaction of plexin-C1 function. Overall the study is adequately executed, using some nice experimental strategies, and it is concisely presented/well written. It should have major impact on the field as this is the first structure of the entire ectodomain of plexin-C1 by cryo-EM at a good resolution. By contrast to an earlier structure of an ectodomain fragment, bound to the same virus derived ligand (ref. 11), the C-terminal region (now present) is bound back to the N-terminal region of the ectodomain, in part by a covalent link/disulphide. This globally links two rod-like structures and appears to ensure that the very C-terminal, membrane proximal region, IPT4, is in relatively close proximity in a dimer. The dimer structure as resolved, then, appears to be held together at the N-terminal end by the dimeric ligand and at the C-terminal end by the transmembrane region of the full-length plexin-C1 protein used. The structural role of several of the inter-domain contacts are backed up by site directed mutagenesis, disrupting them in conjunction with a simple cell based functional assay (cell collapse). Overall this is a novel and significant finding especially because it contrasts a crystal/cryoEM structure presented for the A-family of plexins, which have more domains in the extracellular region and show a ring-like structure in the apo-form.

This reviewer has three major concerns about the interpretation of the study, two of which need to be addressed by additional experiments.

- 1) The disulphide formed in plexin-C1 between Cys412 of Blade 6 of the SEMA domain and Cys703 in the C-C' loop of IPT2 is likely the essential stabilizer of the interdomain interactions. The statement in line 308 that the disulphide alone is not sufficient assumes that it is still formed in the triple mutant that is disrupting the other/Cys surrounding interactions. Maybe it is not formed kinetically under the conditions used? Whether or not this is the case should be shown by additional structural work (see comment on line 308 below). The discussion of this disulphide needs to be expanded...e.g. whether there could be an equivalent in other plexins and whether the disulphide would be found in non-mammalian species. If not, this might suggest a different mechanism/that some other interactions might compensate. The concern is a possible overinterpretation of the finding as the disulphide may only have formed under the purification conditions used, i.e. in a native-cellular context additional/different carbohydrates may block its formation. Alternatively, if no structure for the mutants can be obtained, one could use the extracellular fragment with a GFP tag at the C-termini to detect FRET when the interdomain interactions, either the disulphide/interactions between SEMA and IPT2 and/or the interactions between IPT4 bring the C-termini together (this also relates to the second major point).
- 2) The second major contributor for the structure to turn into two rod-like regions which turn back to bring the C-terminal IPT4 region together near the membrane, is likely to be the dimerization of the TM as well as of the intracellular regions. However, a major caveat to this is the use of the peptidisc which will likely clamp together two plexin proteins and thus bring the IPT4 regions into proximity. This could be addressed in several ways - one would be to make a chimera of the plexin-C1 extracellular domain with a TM-intracellular protein known to remain monomeric in membranes or to make mutations in the residues that were predicted to hold the plexin-C1 TM

region together as a dimer (see doi: 10.1371/journal.pone.0121513 – which in any case should be cited in line 176 of the manuscript and then later in the discussion lines 359-371). Part of this issue is mitigated by mutating in glycosylation sites, described in line 315, although in absence of further experiments/structural data, one would worry that rather than disturbing the IPT4-IPT4 interaction, these may interfere with other interactions, such as the orientation of IPT4 with respect to the membrane or the other domains. See also criticism of line 232 below.

3) The title of the paper needs to be changed from “cryo-EM structure of the full length complex” to “cryo-EM structure reveals new inter-domain interactions in the extracellular region of the Plexin-C1/A39R complex”. Then in the abstract it should be mentioned that the partial structure was obtained with the full length protein. The issue is that many readers who are not in the plexin field will likely only notice the title of the paper and will infer that the whole length structure of a plexin has been resolved by cryo-EM, which clearly is not the case. This may hinder future efforts at publication and/or funding.

Below follow several more issues of moderate to minor importance that should be addressed in a revision. These are given in order of occurrence in the manuscript/with line numbers.

Title and line 26: see point 3 above.

Line 31: see points 1 and 2. Again, the covalent and TM dimerization may be artificial...wording should be softened.

Line 42: it is fine to cite their own review and that of another structural biology lab., but a third review on the same topic (doi: 10.1007/s00018-012-1019-0) should be cited as well (also, since it has gathered a larger number of citations than the other two).

Line 82: The authors should emphasize and later discuss that the plexin-A ring structure was determined for plexin-A family members in absence of ligand. The structure of the ligand bound plexins possibly of the A, B and D-families may include a similar kink that turns the structures around (esp. in case of D-family whose extracellular domain also lacks several PSI/ITPs).

Line 98: see as part of point 2 above, the reference to an all-atom prediction and extensive molecular dynamics refinement of the plexin-C1 TM region should be cited here.

Line 147: change “membrane proximal domains” to “the C-terminal extracellular membrane proximal domain, IPT4 and the TM region” (as the former statement would seem to include the cytoplasmic juxtamembrane region which is not seen).

Line 127: the improvement of the resolution due to the application of symmetry is relatively slight, 2.9 vs. 3.1 Angstrom for the most resolved region. The authors should comment extensively, perhaps in the supplement, how different the non-symmetric structures are. A diagram that gives resolution mapped onto the structure should be shown for the refinement of the dimer also without use of symmetry restraint.

Line 146 and Fig. S5 – the figure has almost no discernable difference/difficult to see. The authors should show some subregions in atomic detail in an expanded Fig. S5 where differences are seen and comment.

Line 232. While the two IPT4's may not make direct contact (here esp. the structure without symmetry restraints would be important) some analysis should be done on the actual distance between modeled sidechains (i.e. 38 Angstrom between two tyrosines seems rather large: what is the closest approach?...could a single water molecule or ion bridge important interactions?) In order to strengthen the paper on this issue of possible IPT4 stabilizing interactions (point 2 above), if such bridging interactions are likely, residue mutations should be made to eliminate them (again, adding big sugar groups is a rather blunt tool).

Line 263: The finding of near identical binding affinity for the ligand between truncated and full length extracellular region plexin-C1 (ref. 11) seems to suggest that the chain reversal (disulphide) which brings the two IPT4 domains together does not lead to an easier/ligand binding

beneficial dimerization of plexin. Thus the importance of the new inter-domain interactions for ligand binding is surprisingly negligible. Need to mention this point more clearly.

Line 308: The statement assumes that the C412/C703 disulphide still forms in presence of these mutations. This is an important issue (see point 1) above and should be resolved by a structural analysis of the mutant protein (by cryo-EM or SASX etc.).

Line 330. plexin-D1 also has fewer extracellular domains.

Line 625/Table 1. It seems very implausible that the previous plexin-C1/A39R complex structure, ref. 11, was not used for model building. In fact that is what it says in the Materials and Methods and thus belongs in the table.

Line 640 Fig. 2 B label PSI & IPT domains in the plexin-A structures. It would seem that at least in some structures in plexin-A interactions between IPT4 and SEMA are seen....how does this compare with the interactions seen for plexin-C1. Please comment.

Line 655: How many residues are there between the last residue resolved in IPT4 and the TM region?

Line 633 – make reference to Fig. S4 (see below line 740).

Line 691: Fig.6 legend, emphasize that all the plexin-A structures were inferred from those in absence of ligand/not active.

Line 707: Fig. S1 – indicate which fractions were used for the cryo-EM samples.

Line 720. Fig. S2 A – hard to see particles (particals misspelled throughout fig.). Please indicate, e.g. circle some. Fig. S2 C – give numbers and percentages and resolution of the two last classes on right. Explain more the use of C1 and C2 symmetry and clarify flow-chart (number of particles seems to down and then up again).

Line 728, Fig. S3 show Euler angle distribution for reconstructions.

Line 740. Show in same orientations as Fig. 4 and label key residue sidechains by their number.

Line 753: Fig. S6, how were the domain limits defined? give source or ref. – indicate which regions are modeled in the structure by line with another color.

Reviewer #3 (Remarks to the Author):

Kuo et al., describe the first structural work on a full-length version of the transmembrane receptor plexin, i.e. PlexinC1 in complex with the a viral semaphorin-ligand mimic A39R. While the structure of a much smaller part of the PlexinC1 receptor in complex with A39R had been described previously, the new cryo-EM structure described here reveals the relationship between the previously unresolved part of the PlexinC1 extracellular region and the dimerization by A39R. The structural data fits with a hypothetical model that had been derived for PlexinA-Semaphorin6 from structures of the PlexinA full extracellular region and structures of smaller segments of PlexinA2 in complex with Semaphorin6A ligand. Kuo et al. experimentally show, and describe clearly, that this previous model is also used by PlexinC1. Nonetheless, substantial differences are apparent, and these are described well in the manuscript.

The structural biology, by single particle cryo-EM, is done to a very high standard. This is one of the first detailed structures of a full-length type I transmembrane protein. It is very unfortunate that the transmembrane part and the cytosolic segment are not resolved in the data, most likely due to flexibility with respect to the extracellular part. This is, however, not surprising, as a detailed structure, that reveals both the extracellular as well as the cytosolic segment a in type I

transmembrane protein has not been described yet for any sample (as indicated by Kuo et al.). The COS-cell collapse assay of mutant versions of PlexinC1 are very appropriate and verify the structural findings.

I have only minor comments:

Line 222: Could the authors perhaps indicate sequence identity numbers for the IPT domains of PlexinC1 with the relevant PlexinA paralogs?

Line 506: Could the authors indicate which parts are omitted from the modelled coordinates, in particular for the IPT1, PSI2, IPT3 and IPT4 domains?

The resolution for the IPT1, IPT3 and IPT4 domains in the density maps shown in suppl. Fig 4, seems to be much lower compared to the rest of the structure. What is the resolution of each of these domains individually? Could the authors indicate if any care was taken to stabilize the refinement for these parts, e.g. secondary structure restraints, modelling based on homology to PlexinA IPT structures, etc?

We greatly appreciate the overall positive comments and constructive critiques of the three reviewers. The following are point-by-point responses to the specific questions.

Reviewers' comments:

Reviewer #1 (Remarks to the Author):

Plexin family proteins represent major class of signaling receptors for semaphorin ligands, which play important roles in various cellular events implicated in normal development as well as diseases. Plexin-mediated signaling requires activation of its GAP activity located in the intracellular region, generally believed to be accomplished by dimerization upon the engagement of dimeric semaphorin ligand at the extracellular face. However, current "plexin activation model" lacks structural and mechanistic precision due to the fact that it is based on the structural information obtained for various partial fragments of both ligands and receptors, as in the case for virtually all single-pass membrane receptor systems.

In this paper, Kuo et al made a huge step toward the complete understanding of the plexin activation on the membrane, by reporting a 3.1Å cryo-EM structure of the entire plexinC1 polypeptide in complex with a semaphorin surrogate ligand A39R. Although they could not build models for the TM and the cytoplasmic GAP domains, they succeeded in visualizing nearly all ectodomain regions for plexin C1 and bound ligand, presenting the global architecture of the 2:2 plexin-ligand complex formed at the extracellular side for the first time. The structure revealed that the two TM domains are indeed in close proximity, supporting the hypothesis that the ligand-induced ectodomain dimerization of plexin leads to the GAP domain dimerization. They also discovered unexpected intra-molecular interface between the sema domain and the "bottom" of the IPT2 domain, which makes the plexin C1 ectodomain highly bent and compact, despite the presence of several Ig-like modules in its "stalk". Apart from the new information related to the plexin signaling mechanisms, this paper also offers useful information regarding the important technical advancement to the field, which is the use of peptidisc approach. Despite extensive efforts by numerous experts in the field, we still do not have atomic resolution structures of single-pass membrane proteins with large ectodomain. This is in contrast to the multi-pass membrane proteins, of which high resolution structures by either cryo-EM or crystallography are reported almost weekly. Peptidisc approach is much less popular than other reconstitution methodologies including nanodisc, so the result by Kuo et al may spark new wave of investigation of single-TM protein reconstitution by using this relatively new technology. Overall, I think this fine paper should be published without unnecessary delays. There are few specific comments/requests I would like to make.

1. Disulfide bridge connecting Cys412 and Cys703

I have a concern about the authenticity of this disulfide. Although this assignment might be consistent with the sequence conservation pattern among different plexins (should be nice to have sequence alignment figure here), the authors do not show experimental evidence to back up this notion. At least very clear density map near this disulfide must be shown, but even then the conclusion must be drawn very carefully considering the low resolution of the data. My concern stems from the puzzling data that charge-neutralizing mutations at the interface alone (i.e., D396A/E422A and K728A) can lead to the disrupted signaling phenotype (Fig. 5). The C412-C703 disulfide would covalently fix the sema-IPT2 interface and hence any mutations at the interface (particularly when they are Ala mutations) should not be able to affect the conformation. The authenticity of this kind of long-range disulfide bond is extremely important to correctly understand the conformational regulation of a receptor, so I urge authors to address this point more rigorously. A limited protease digestion on the full-length (or recombinant soluble ectodomain) plexinC1 could clarify this point, because the disordered PSI2-IPT2 linker region should be susceptible to such digestion. If authors cannot give more supporting evidence for the disulfide, they should mention that

the disulfide formation is one of the potential explanations among other possibilities.

The density map of the disulfide bond is now included in Supplemental Figure 5, lower right panel. The suggested experiment for directly assessing the inter-domain disulfide by this reviewer is a great idea. We expressed and purified the extracellular region of human PlexinC1 and attempted the experiments with trypsin digestion. Unfortunately, the protein was cleaved into many pieces, owing to the many lysine and arginine residues present in the protein. As an alternative, we inserted an HRV 3C protease (a.k.a. PreScission protease) site in the linker between PSI2 and IPT2. Treatment of this protein with 3C protease led to two bands under reducing conditions, corresponding to the N- and C-terminal domains respectively. In contrast, the treated protein remains as one band under non-reducing conditions. These results strongly support the notion that the N- and C-terminal segments are linked by an inter-domain disulfide bond. These results are included as Supplemental figure 7.

Regarding the puzzling results (which is also raised by reviewer 2), there are two potential explanations. The first is that the mutations of residues involved in the un-covalent interactions between the Sema domain and IPT2 may destabilize the relative orientation between the two domains, leading to reduced efficiency in the formation of the disulfide bond. The second is that the disulfide bond alone is not sufficient to fix the orientation of the membrane proximal domains, leading to impaired signaling. We have softened the language in this paragraph to reflect this ambiguity.

2. IPT domain conformations and arrangements

The authors state that "the curved rod shape of the IPT2-4 in PlexinC1 does not resemble any of the three consecutive IPT domains in class A plexins". This notion is central to their point of argument regarding the unique IPT domain conformation (and signaling mechanism) of PlexinC1, but unfortunately the current figures cannot convey this message. I suggest authors to make superpositions of the consecutive as well as individual IPT domains to help readers to appreciate similarity and difference among various IPT domains. Also, a supplementary figure showing structurally-aligned IPT domain sequences including other plexin classes, with secondary structure elements clearly labeled, would be very informative to readers.

This is a great suggestion. As indicated in the original manuscript, from the structural point of view, IPT2-4 of PlexinC1 and IPT4-6 of PlexinA1 can be considered equivalent as they are the most membrane proximal domains that link to the transmembrane region of the proteins. Therefore, we now include a structural superimposition of IPT2-4 of PlexinC1 and IPT4-6 of PlexinA1 (Figure 2d), which shows that the shapes of the two modules are very different.

We realize that the statements in this paragraph of the original manuscript failed to convey the point that the distinct shape of IPT2-4 in PlexinC1 alone is not sufficient to determine the position of the transmembrane region and signaling mechanism. The position of IPT2-4 relative to the Sema domain is also critical. We have revised this paragraph to clarify this point.

The sequence identities between the individual PlexinC1 IPT domains and those of class A plexins are very low (in the range of 9-25%). Based merely on sequence similarity, it is not even clear which IPT domains from different plexin classes correspond to each other. We therefore decided not to show the alignments of the IPT domains of different plexin classes. We agree that the labelling the secondary structural elements of the IPT domains is informative to readers, which is included in the revised Supplemental figure 6.

3. peptidisc reconstitution

In general, the method section is written with sufficient details. The EM analysis part is well performed, applying some of the analytical approaches that are at the forefront of the field. Unfortunately, the authors give only limited information regarding their preparation of the peptidisc

reconstitution, which is probably the most wanted information from the membrane protein researchers. I urge authors to expand this section and describe fully the content of the peptidisc with actual sequence of the peptide used, so that readers do not have to go back and forth between the ref 15 and the company's URL.

(end of comments)

We agree and have included the sequence and more details on the reconstitution of peptidisc in the method section.

Reviewer #2 (Remarks to the Author):

The paper by Bai, Zhang and coworkers claims to reveal inter-domain interactions critical for the ligand induced interaction of plexin-C1 function. Overall the study is adequately executed, using some nice experimental strategies, and it is concisely presented/well written. It should have major impact on the field as this is the first structure of the entire ectodomain of plexin-C1 by cryo-EM at a good resolution. By contrast to an earlier structure of an ectodomain fragment, bound to the same virus derived ligand (ref. 11), the C-terminal region (now present) is bound back to the N-terminal region of the ectodomain, in part by a covalent link/disulphide. This globally links two rod-like structures and appears to ensure that the very C-terminal, membrane proximal region, IPT4, is in relatively close proximity in a dimer. The dimer structure as resolved, then, appears to be held together at the N-terminal end by the dimeric ligand and at the C-terminal end by the transmembrane region of the full-length plexin-C1 protein used. The structural role of several of the inter-domain contacts are backed up by site directed mutagenesis, disrupting them in conjunction with a simple cell based functional assay (cell collapse). Overall this is a novel and significant finding especially because it contrasts a crystal/cryoEM structure presented for the A-family of plexins, which have more domains in the extracellular region and show a ring-like structure in the apo-form.

This reviewer has three major concerns about the interpretation of the study, two of which need to be addressed by additional experiments.

1) The disulphide formed in plexin-C1 between Cys412 of Blade 6 of the SEMA domain and Cys703 in the C-C' loop of IPT2 is likely the essential stabilizer of the interdomain interactions. The statement in line 308 that the disulphide alone is not sufficient assumes that it is still formed in the triple mutant that is disrupting the other/Cys surrounding interactions. Maybe it is not formed kinetically under the conditions used? Whether or not this is the case should be shown by additional structural work (see comment on line 308 below). The discussion of this disulphide needs to be expanded...e.g. whether there could be an equivalent in other plexins and whether the disulphide would be found in non-mammalian species. If not, this might suggest a different mechanism/that some other interactions might compensate. The concern is a possible overinterpretation of the finding as the disulphide may only have formed under the purification conditions used, i.e. in a native-cellular context additional/different carbohydrates may block its formation. Alternatively, if no structure for the mutants can be obtained, one could use the extracellular fragment with a GFP tag at the C-termini to detect FRET when the interdomain interactions, either the disulphide/interactions between SEMA and IPT2 and/or the interactions between IPT4 bring the C-termini together (this also relates to the second major point).

We agree that this is an important point, which is also raised by Reviewer 1. These two cysteine residues are indeed conserved in all the PlexinC1 sequences that we found, including those from birds (chicken and zebra finch) and zebrafish, strongly suggesting that this disulfide is a conserved mechanism for stabilizing the relative orientation between the Sema and membrane proximal domains. We agree that the assumption that the mutant still form the disulfide is unwarranted. We have revised the text of this part to reflect this ambiguity.

We respectfully argue that structural analyses of mutants regarding the disulfide issue would be too time consuming and therefore not feasible at the current stage. To further investigate the formation of the disulfide bond, as suggested by Reviewer 1, we have expressed the extracellular region of PlexinC1 with an HRV 3C protease site in the PSI2-IPT2 linker. We carried out 3C protease digestion and analyses with reducing and non-reducing gels. The results, shown in Supplemental figure 7, further support the formation of the inter-domain disulfide bond. We have also included the density map of the disulfide bond in Supplemental figure 5, lower right panel.

2) The second major contributor for the structure to turn into two rod-like regions which turn back to bring the C-terminal IPT4 region together near the membrane, is likely to be the dimerization of the TM as well as of the intracellular regions. However, a major caveat to this is the use of the peptidisc which will likely clamp together two plexin proteins and thus bring the IPT4 regions into proximity. This could be addressed in several ways - one would be to make a chimera of the plexin-C1 extracellular domain with a TM-intracellular protein known to remain monomeric in membranes or to make mutations in the residues that were predicted to hold the plexin-C1 TM region together as a dimer (see doi: 10.1371/journal.pone.0121513 – which in any case should be cited in line 176 of the manuscript and then later in the discussion lines 359-371). Part of this issue is mitigated by mutating in glycosylation sites, described in line 315, although in absence of further experiments/structural data, one would worry that rather than disturbing the IPT4-IPT4 interaction, these may interfere with other interactions, such as the orientation of IPT4 with respect to the membrane or the other domains. See also criticism of line 232 below.

We believe that it is highly unlikely that the shape of the dimeric complex is distorted by the peptidisc. During our optimization of the complex samples for cryo-EM, we collected data of the complex in amphipol, a polymer that is also often used to replace detergent for stabilizing transmembrane region in cryo-EM structural analyses. The reconstruction we got from this data did not reach a resolution as high as the peptidisc sample, but showed that the conformation of the complex is virtually identical to the structure shown in the manuscript (Left panel below). In the amphipol sample, we found one 3D class of particles that contains a dimeric A39R bound to only one PlexinC1 molecule (perhaps due to sub-stoichiometric PlexinC1 when mixing the two proteins for complex formation). PlexinC1 in this class adopts a very similar conformation as that in the 2:2 complex (Right panel below). These observations together suggest that the relatively rigid shape of the PlexinC1 extracellular region and the binding mode with A39R together determine the close juxtaposition of the membrane proximal domains. While the peptidisc may have helped stabilize this pre-existing conformation of the complex, it did not dictate this conformation. We decided not to present these data in the paper because the resolution is lower, and the information is largely redundant.

That said, we agree that this is an important point and therefore carried out additional mutational analyses to test our model. The new data is now included in Figure 5, and we have added a few sentences at the end of the result section to describe these results. In essence, we found that a 4-residue deletion in the linker between the IPT4 and transmembrane region led to loss of cell collapse, while a 4-residue insertion reduced collapse. These results are consistent with the model where the two transmembrane regions need to come close for signaling, as the deletion make the linker too short to bridge the 38Å gap between the two monomers, while the insert can be tolerated to some extent.

We are sorry for omitting the important computational paper on the plexin transmembrane region in the reference. It is corrected now.

3) The title of the paper needs to be changed from “cryo-EM structure of the full length complex” to “cryo-EM structure reveals new inter-domain interactions in the extracellular region of the Plexin-C1/A39R complex”. Then in the abstract it should be mentioned that the partial structure was obtained with the full length protein. The issue is that many readers who are not in the plexin field will likely only notice the title of the paper and will infer that the whole length structure of a plexin has been resolved by cryo-EM, which clearly is not the case. This may hinder future efforts at publication and/or funding.

As suggested, we have removed “full-length” in the title. The rest of the manuscript has been carefully worded to not overstate the achievement of the work by making it clear that the transmembrane and intracellular domains are not resolved in the current structure.

Below follow several more issues of moderate to minor importance that should be addressed in a revision. These are given in order of occurrence in the manuscript/with line numbers.

Title and line 26: see point 3 above.

Line 31: see points 1 and 2. Again, the covalent and TM dimerization may be artificial...wording should be softened.

Please see the response to Major point 1 above.

Line 42: it is fine to cite their own review and that of another structural biology lab., but a third review on the same topic (doi: 10.1007/s00018-012-1019-0) should be cited as well (also, since it has gathered a larger number of citations than the other two).

Sorry for this oversight. This reference has been added.

Line 82: The authors should emphasize and later discuss that the plexin-A ring structure was determined for plexin-A family members in absence of ligand. The structure of the ligand bound plexins possibly of the A, B and D-families may include a similar kink that turns the structures around (esp. in case of D-family whose extracellular domain also lacks several PSI/ITPs).

We agree and now include a few sentences in the figure legend of Figure 6 to make it clear that the ring-like structures of class A plexin were in the apo-state, and the overall complex is a model, rather than actual crystal structures. We believe that the descriptions in line 82 also clearly state this fact. We double checked and found that PlexinD1 has the same domain structure as PlexinA.

Line 98: see as part of point 2 above, the reference to an all-atom prediction and extensive molecular dynamics refinement of the plexin-C1 TM region should be cited here.

Done as suggested.

Line 147: change “membrane proximal domains” to “the C-terminal extracellular membrane proximal domain, IPT4 and the TM region” (as the former statement would seem to include the cytoplasmic juxtamembrane region which is not seen).

This is a great suggestion. We have added “extracellular” in this sentence to avoid the potential confusion.

Line 127: the improvement of the resolution due to the application of symmetry is relatively slight, 2.9 vs. 3.1 Angstrom for the most resolved region. The authors should comment extensively, perhaps in the supplement, how different the non-symmetric structures are. A diagram that gives resolution mapped onto the structure should be shown for the refinement of the dimer also without use of symmetry restraint.

The improvement of the overall resolution was not dramatic, but did help the interpretation of the map for model building, particularly for the IPT domains. The difference described is not due to lack of symmetry in the dimeric structure. It is probably better considered as small hinge motions of the two halves of the complex around the dimeric interface of A39R, which does not break the symmetry but leads to slightly different conformations among different particles. The consequence of this is that regions far away from the hinge are less well aligned in the 3D reconstruction, leading to lower local resolution that are far away from the 2-fold symmetry axis. The symmetry expansion and focused refinement procedures in Relion were specifically designed to deal with situations like this. It improves the resolution because it allows each half of the dimeric complex to be considered independently and better aligned, reducing the detrimental effects of the hinge motion. The half-model and full dimeric model are therefore very similar in conformation. We have added a few sentences in the method to further clarify this.

Line 146 and Fig. S5 – the figure has almost no discernable difference/difficult to see. The authors should show some subregions in atomic detail in an expanded Fig. S5 where differences are seen and comment.

The two structures are indeed very similar in this part, and we could not identify any differences of interest. The intent of this figure is to show this similarity. We have enlarged the panels to make it clearer.

Line 232. While the two IPT4's may not make direct contact (here esp. the structure without symmetry restraints would be important) some analysis should be done on the actual distance between modeled sidechains (i.e. 38 Angstrom between two tyrosines seems rather large: what is the closest approach?...could a single water molecule or ion bridge important interactions?) In order to strengthen the paper on this issue of possible IPT4 stabilizing interactions (point 2 above), if such bridging interactions are likely, residue mutations should be made to eliminate them (again, adding big sugar groups is a rather blunt tool).

See the response to point 2 above, we believe that our new data on the deletion and insertion in the linker region at least partially address this point. It is difficult at this point to speculate whether the linker regions from the two plexin molecules make any specific interactions to stabilize the dimeric complex because they are disordered.

Line 263: The finding of near identical binding affinity for the ligand between truncated and full length extracellular region plexin-C1 (ref. 11) seems to suggest that the chain reversal (disulphide) which brings the two IPT4 domains together does not lead to an easier/ligand binding beneficial dimerization of plexin. Thus the importance of the new inter-domain interactions for ligand binding is surprisingly negligible. Need to mention this point more clearly.

We are a bit puzzled by this comment. We do not indicate that the inter-domain interaction has an effect on ligand binding, or vice versa. In fact, we cited this paper to make the point that there is no allosteric coupling (such as a long-range conformational change that requires energy from the ligand binding) between the sema domain and the membrane proximal domains in PlexinC1. It follows that, in the absence of the transmembrane region and intracellular region, the binding of the ligand to the

isolated Sema domain or intact extracellular region of PlexinC1 is expected to be the same. We have added a sentence to clarify this point.

The key point is that the inter-domain interaction is essential to determine the orientation of the membrane proximal domains relative to the sema domain, which is required to place the two copies of the membrane proximal domains into proximity in the dimeric complex to induce the dimerization of the transmembrane and intracellular region.

Line 308: The statement assumes that the C412/C703 disulphide still forms in presence of these mutations. This is an important issue (see point 1) above and should be resolved by a structural analysis of the mutant protein (by cryo-EM or SASX etc.).

We agree and please see the response to point 1 above.

Line 330. plexin-D1 also has fewer extracellular domains.

We double checked and found that PlexinD1 has the same domain structure as class A plexins. Did we miss something here?

Line 625/Table 1. It seems very implausible that the previous plexin-C1/A39R complex structure, ref. 11, was not used for model building. In fact that is what it says in the Materials and Methods and thus belongs in the table.

Thanks for pointing out this error. It has been corrected.

Line 640 Fig. 2 B label PSI & IPT domains in the plexin-A structures. It would seem that at least in some structures in plexin-A interactions between IPT4 and SEMA are seen....how does this compare with the interactions seen for plexin-C1. Please comment.

The domains are now labelled. As pointed out by this reviewer as well as in the main text of the manuscript, in one of the PlexinA1 structures (PDB ID: 5I56) there seem to be some loose interactions between the Sema domain and IPT5. The interface is however very small, and not present in the two other structures of the same protein. It is not clear at this point whether this interface plays a role in the activation of PlexinA1.

Line 655: How many residues are there between the last residue resolved in IPT4 and the TM region?

There are nine residues from each Plexin molecule (as stated in the text; we also include this information in the revised figure now). As mentioned above, we have also tested deletion and insertion mutants in this linker.

Line 633 – make reference to Fig. S4 (see below line 740).

Done

Line 691: Fig.6 legend, emphasize that all the plexin-A structures were inferred from those in absence of ligand/not active.

Done

Line 707: Fig. S1 – indicate which fractions were used for the cryo-EM samples.

Done

Line 720. Fig. S2 A – hard to see particles (particals misspelled throughout fig.). Please indicate, e.g. circle some. Fig. S2 C – give numbers and percentages and resolution of the two last classes on right. Explain more the use of C1 and C2 symmetry and clarify flow-chart (number of particles seems to down and then up again).

Thanks for pointing out the typos, which have been corrected. A new raw image with some particles circled has been included, which makes it easier to see particles. The plexin complex particles are indeed difficult to see due to the highly irregular shape, in contrast to most multi-pass transmembrane proteins such as transmembrane ion channels which have a typical shape (large disk shape of the membrane part plus soluble domain blob) that is easy to identify.

Particle number increase because symmetry expansion leads to doubling of the particles, as each half of the original dimeric particles is counted as one particle. We do not show resolution of individual 3D classes because they are not calculated according to the gold-standard FSC method and therefore could be misleading. When refining the intact 2:2 complex, C2 symmetry is used because the complex has such symmetry. In focused refinement, the particles were symmetry expanded (hence the 2-fold increase of the particle number) and subtracted to contain only half of the complex, which does not have the C2 symmetry and therefore refined with C1 (no symmetry).

Line 728, Fig. S3 show Euler angle distribution for reconstructions.

Done. We only included this for the dimeric reconstruction, because the focused reconstruction of the half complex is based on the dimeric data and therefore the distribution is identical.

Line 740. Show in same orientations as Fig. 4 and label key residue sidechains by their number.

As suggested, some residues are labelled now in these figures. In addition, we include the density of the disulfide bond, where the two cysteine residues are labelled.

Line 753: Fig. S6, how were the domain limits defined? give source or ref. – indicate which regions are modeled in the structure by line with another color.

Domain limits are defined based on the structure in combination with secondary structure prediction. This information has been added in the figure legend. We have also added the secondary structure elements of the IPT domains. We chose not to highlight regions that are modelled or not modelled because the figure has become somewhat cluttered/confusing. We added a few sentences in the method section to clarify which residues are omitted in the model. The structure coordinates will be released to the public, which will be more informative for readers to find out which parts of the protein are modelled or missing.

Reviewer #3 (Remarks to the Author):

Kuo et al., describe the first structural work on a full-length version of the transmembrane receptor plexin, i.e. PlexinC1 in complex with the a viral semaphorin-ligand mimic A39R. While the structure of a much smaller part of the PlexinC1 receptor in complex with A39R had been described previously, the new cryo-EM structure described here reveals the relationship between the previously unresolved part of the PlexinC1 extracellular region and the dimerization by A39R. The structural data fits with a hypothetical model that had been derived for PlexinA-Semaphorin6 from structures of the PlexinA full extracellular region and structures of smaller segments of PlexinA2 in complex with Semaphorin6A ligand. Kuo et al. experimentally show, and describe clearly, that this previous model is also used by PlexinC1. Nonetheless, substantial differences are apparent, and these are described well in the manuscript.

The structural biology, by single particle cryo-EM, is done to a very high standard. This is one of the first detailed structures of a full-length type I transmembrane protein. It is very unfortunate that the transmembrane part and the cytosolic segment are not resolved in the data, most likely due to flexibility with respect to the extracellular part. This is, however, not surprising, as a detailed structure, that reveals both the extracellular as well as the cytosolic segment a in type I transmembrane protein has not been described yet for any sample (as indicated by Kuo et al.). The COS-cell collapse assay of mutant versions of PlexinC1 are very appropriate and verify the structural findings.

I have only minor comments:

Line 222: Could the authors perhaps indicate sequence identity numbers for the IPT domains of PlexinC1 with the relevant PlexinA paralogs?

The sequence identity of different IPT domains of human PlexinC1 versus those in human PlexinA1 is within the range of 9-25%. We have included this information in the revised version as suggested.

Line 506: Could the authors indicate which parts are omitted from the modelled coordinates, in particular for the IPT1, PSI2, IPT3 and IPT4 domains?

As suggested, this information is included in the method section of the revised manuscript.

The resolution for the IPT1, IPT3 and IPT4 domains in the density maps shown in suppl. Fig 4, seems to be much lower compared to the rest of the structure. What is the resolution of each of these domains individually? Could the authors indicate if any care was taken to stabilize the refinement for these parts, e.g. secondary structure restraints, modelling based on homology to PlexinA IPT structures, etc?

The resolution of these domains is indeed low, as indicated by the local resolution map shown in the supplemental figure. We did use secondary structure restraints and modelling based on homology to IPT structures of class A plexins. Bulky glycosylation groups also helped determining the sequence register. This information is now included in the method section.

REVIEWERS' COMMENTS:

Reviewer #1 (Remarks to the Author):

The authors have addressed all the issues raised by this reviewer by adding new experimental results and revising the figures/texts. I have no further requests and comments and the paper seems ready to be published.

Reviewer #2 (Remarks to the Author):

Overall the paper has been adequately revised addressing the majority of the reviewers criticisms. The authors performed and included two key experiments. The one point that should still be made however is that the peptidisc is a very poor representative of a lipid bilayer in that it does not allow enough space for the transmembrane (TM) regions to diffuse freely, certainly not over a long distance. In absence of an example where the extracellular regions form a dimer but TM regions are solvated by separate peptidiscs, it is reasonable to assume that as soon as two extracellular regions bring them in some proximity the peptidisc will have a considerable effect in bringing the TM regions close together. This caveat should be mentioned in the discussion.

Reviewer #2 (Remarks to the Author):

Overall the paper has been adequately revised addressing the majority of the reviewers criticisms. The authors performed and included two key experiments. The one point that should still be made however is that the peptidisc is a very poor representative of a lipid bilayer in that it does not allow enough space for the transmembrane (TM) regions to diffuse freely, certainly not over a long distance. In absence of an example where the extracellular regions form a dimer but TM regions are solvated by separate peptidiscs, it is reasonable to assume that as soon as two extracellular regions bring them in some proximity the peptidisc will have a considerable effect in bringing the TM regions close together. This caveat should be mentioned in the discussion.

We have added the following sentence in the discussion to address this point: “In addition, it remains possible that the peptidisc might have imposed some restraints on the conformation of the PlexinC1 transmembrane region.”